# Graph-KV: Breaking Sequence via Injecting Structural Biases into Large Language Models

**Haoyu Wang[1]\*, Peihao Wang[2], Mufei Li[1], Shikun Liu[1], Siqi Miao[1], Zhangyang Wang[2], Pan Li[1]\***

[1] Georgia Institute of Technology [2] The University of Texas at Austin

## Abstract

Modern large language models (LLMs) are inherently auto-regressive, requiring input to be serialized into flat sequences regardless of their structural dependencies. This serialization hinders the model's ability to leverage structural inductive biases, especially in tasks such as retrieval-augmented generation (RAG) and reasoning on data with native graph structures, where inter-segment dependencies are crucial. We introduce Graph-KV with the potential to overcome this limitation. Graph-KV leverages the KV-cache of text segments as condensed representations and governs their interaction through structural inductive biases. In this framework, "target" segments selectively attend only to the KV-caches of their designated "source" segments, rather than all preceding segments in a serialized sequence. This approach induces a graph-structured block mask, sparsifying attention and enabling a message-passing-like step within the LLM. Furthermore, strategically allocated positional encodings for source and target segments reduce positional bias and context window consumption. We evaluate Graph-KV across three scenarios: (1) seven RAG benchmarks spanning direct inference, multi-hop reasoning, and long-document understanding; (2) ARXIV-QA, a novel academic paper QA task with full-text scientific papers structured as citation ego-graphs; and (3) paper topic classification within a citation network. By effectively reducing positional bias and harnessing structural inductive biases, Graph-KV substantially outperforms baselines, including standard costly sequential encoding, across various settings. Code and the ARXIV-QA data are publicly available at `https://github.com/Graph-COM/GraphKV`.

## 1 Introduction

Modern large language models (LLMs) [1, 50, 2], despite their notable successes, are fundamentally auto-regressive. This characteristic, as a consequence of their training approaches [51, 41], necessitates the serialization of information for processing. Consequently, all input, regardless of its intrinsic structure or complex dependencies, such as order-insensitivity, temporal or logical relationships, must be flattened into an ordered sequence. This forced serialization can be suboptimal and may introduce a sequential bias, potentially hindering the LLM's ability to fully leverage these internal relationships.

For example, in retrieval-augmented generation (RAG) [29, 11, 67, 30], retrieved text segments, which may lack a linear order or possess complex, non-linear interdependencies, must still be artificially serialized, which can limit effective multi-hop reasoning [17, 62, 44, 49] and introduce positional biases [58, 15, 66]. Similarly, processing data with native graph structures, such as citation networks [20, 14] where citations signify knowledge dependencies, presents challenges. Serializing documents referenced by the same document, for instance, leads to drawbacks including: 1) positional biases that can obscure parallel citation relationships; 2) quadratic computational complexity when

---

\*Correspondence to: haoyu.wang@gatech.edu, panli@gatech.edu.

39th Conference on Neural Information Processing Systems (NeurIPS 2025).

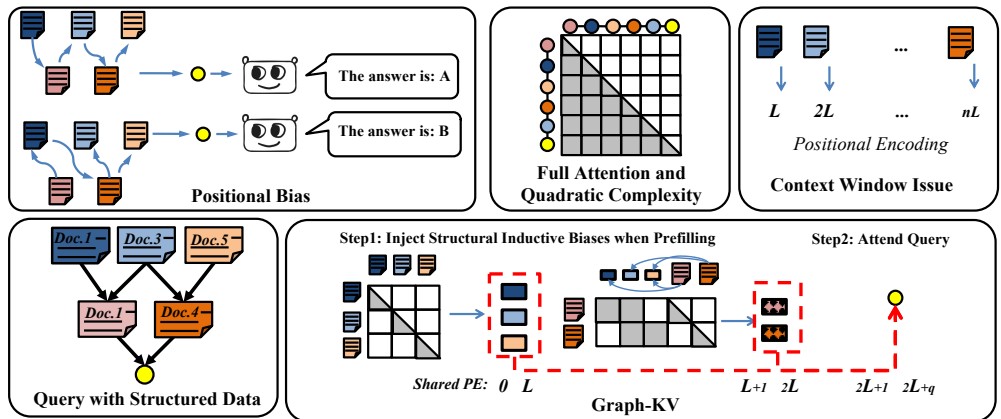

Figure 1: When processing data with inherent structure (**bottom-left**), modern LLMs encounter three challenges due to serialized input reading (**top row**): (1) positional bias, where different serialization orders lead to varied outputs [69];(2) quadratic computational complexity from full attention between all document pairs; and (3) rapid context window consumption, as position indices quickly exceed limits. The **bottom-right** subfigure illustrates Graph-KV. Text chunks are independently encoded into KV caches, where Graph-KV arranges the text chunk of each target text after the KV of their source texts to update their respective KV caches. Notably, all source texts share same positional encoding (PE) range, while all target texts share another, with their position index immediately following that of the source nodes. This approach reduces the PE and context window usage. At query time, the query attends to both the source chunks and the target chunks to perform decoding.

attending to all document pairs; and 3) context window limitations when dealing with numerous references (Fig. 1).

Therefore, a critical question arises: ***How can we align the structural inductive bias of the data with the mechanisms of auto-regressive LLMs, moving beyond simplistic serialization?***

Existing literature has explored mitigating the negative effects of token-serialization, primarily by aiming to eliminate positional bias and enable LLMs to process text segments in a permutation-invariant manner. For instance, Wang et al. [56] proposed to reorder documents based on attention scores computed without positional encodings (PEs); however, the requisite sorting and full attention computations in this method introduce significant computational overhead. Other works prioritize efficient inference by prefilling the key-value (KV) cache of documents independently in parallel [61, 43, 12, 35]. In these approaches, the encoding of documents within the KV-cache often relies on either shared PEs or sorting based on semantic similarity scores from external retrievers. Although these parallel encoding strategies enhance efficiency, they inherently cannot model inter-segment dependencies, let alone native inductive biases within the structured data (as summarized in Table 1).

To address these limitations, we introduce Graph-KV. The core principle of Graph-KV is to treat the KV cache of a given text segment as its condensed information representation and to control its generation using structural inductive biases. Specifically, after initially prefilling the independent KV caches of all text segments, a "target" segment's KV cache is generated by attending only to the KV caches of its "source" text segments, rather than to all segments that merely precede it in the token-serialization sequence. The determination of "source → target" relationships is guided by

| | Long-Context Friendly | Sparse Attention & Efficiency | Free from Positional Biases | Structural Inductive Biases |
|---|---|---|---|---|
| Sequential Encoding | ✗ | ✗ | ✗ | ✗ |
| Promptcache [12] | ✗ | ✓ | ✗ | ✗ |
| PINE [56] | ✗ | ✗ | ✓ | ✗ |
| PCW [43] | ✓ | ✓ | ✓ | ✗ |
| APE [61] | ✓ | ✓ | ✓ | ✗ |
| Block-Attention [35] | ✗ | ✓ | ✗ | ✗ |
| Graph-KV (ours) | ✓ | ✓ | ✓ | ✓ |

Table 1: Comparison among existing approaches. "Long-Context Friendly" refers to avoiding of rapid context window consumption as the number of input text chunks increases. "Free from positional bias" means model predictions remain stable irrespective of the input chunks' placement order.

structural inductive biases tied to either the data or the specific tasks. From another perspective, this approach essentially introduces a graph-structured block mask (Fig. 1) that sparsifies attention computation during KV cache generation, effectively enabling a "message passing through graph" step within the LLM. Moreover, to mitigate inherent positional biases from the LLM, the attention computation imposes shared PEs across the source segments, with the target segment receiving PEs with position indices immediately following its sources. This design substantially reduces context window consumption through shared PEs while preserving structural alignment.

We evaluated Graph-KV across three diverse settings. First, Graph-KV was assessed on seven RAG benchmarks, covering direct inference [22, 25], multi-hop reasoning [49, 17, 44, 62], and long-document understanding [4]. For these tasks, where native graph structures are absent, we introduced a bipartite graph to establish structural bias between text segments. Across all RAG benchmarks, Graph-KV significantly outperformed parallel text encoding baselines. Notably, in multi-hop reasoning tasks, Graph-KV surpassed even sequential reading while maintaining sparse computation. Second, we introduced ARXIV-QA, a novel and challenging task featuring real-world graph biases. In ARXIV-QA, questions are constructed from the full text of a central scientific paper and its linked references, sourced from the arXiv citation network [20]. These questions require probing technical details and understanding both content and citation relationships. On ARXIV-QA, existing efficient parallel text encoding baselines performed poorly, and standard sequential encoding demonstrated severe positional biases. In contrast, Graph-KV exhibited significant robustness, achieving performance comparable to the peak results of sequential encoding (which necessitates optimal document positioning) without displaying such sensitivity. Third, Graph-KV was evaluated on paper topic classification tasks within citation networks, which possess inherent structural biases through citation links. In this setup, LLMs must classify a central paper by analyzing its title, abstract, and potentially hundreds of references. Graph-KV demonstrated significantly superior performance compared to both sequential encoding and parallel text encoding baselines.

## 2    Related Work

**Positional Bias in LLMs.** Large Language Models (LLMs) exhibit positional bias, wherein their performance is adversely affected by the sequential order of input data [69, 55, 71, 46, 18, 66, 33, 15]. This phenomenon is widely believed to stem from the interplay of Positional Embeddings (PEs) [47, 58, 24, 40] and the inherent causal attention mechanism [42]. Although some research indicates that removing PEs from the transformer architecture can enhance LLM generalization to longer context windows [54], the causal attention mechanism itself can still implicitly induce positional biases [24, 16]. Concrete examples of positional bias are evident in RAG, where models often favor information placed at the beginning or end of the context [33, 39], and in in-context learning, where the order of examples significantly impacts outcomes [68, 34]. The tasks investigated in this work necessitate a more explicit capture of structural dependencies, and our findings reveal that naively serializing input exacerbates positional bias in such scenarios.

**Parallel Encoding and Block Attention.** Research has explored techniques to avoid quadratic computational complexity in RAG for generating KV caches for retrieved documents individually and in parallel [21, 43, 12, 61, 72, 64, 70, 63]. PCW [43] initiated this line of study; however, its performance can degrade substantially in many cases due to distribution shifts in the new form of KV caches. APE [61] proposes a fine-tuning-free algorithm that mitigates distribution shifts with parallel encoding by re-scaling attention magnitudes. [72] further trains a small-LM as a scorer to refine the retrieved parallel contexts. Block-Attention [35] demonstrates further performance improvements over these methods, attributed to its more extensive post-training process. However, a common limitation of these parallel processing strategies is their failure to model inter-document dependencies. Graph-KV mitigates this limitation while preserving the efficiency of parallel encoding.

**Modeling structured data with LLMs.** LLMs predominantly process structured data via two main strategies. The first serializes structured information like graphs into natural language formats for model input [5, 10, 38]. However, this method faces scalability issues from quadratic attention complexity and the inherent challenge of accurately verbalizing intricate structural dependencies. As a result, even reasoning over moderately sized text-attributed graphs (e.g., tens of documents, $100k+$ tokens) can be problematic [19, 53]. The second strategy uses adapter modules to project graph data into the LLM's token embedding space [60, 26, 6, 52, 48]. These adapter-based solutions often exhibit limited generalization, largely due to challenges in achieving robust adapter-mediated alignment [7, 32, 31, 73]. Graph-KV offers a distinct, more foundational approach by being the first to directly modify the LLM's attention mechanism for structured data integration.

**The Challenge of Noisy Multi-Hop Reasoning.** Multi-hop reasoning, which demands capturing arbitrary structural dependencies among multiple pieces of information, remains challenging for LLMs with standard sequential encoding. This difficulty is substantially amplified in real-world scenarios where information is non-contiguous and sparsely distributed within noisy, long contexts, leading to N-fold reductions in LLM performance [3, 27, 65]. However, as our experiments demonstrate, with structural inductive biases, Graph-KV can significantly improve the reasoning capabilities of LLMs.

# 3 Methodology

In this section, we introduce Graph-KV. We assume structural inductive biases can be described by a graph connecting text chunks. Such biases might originate natively from the data or be defined based on the tasks. How these are specified for various tasks will be detailed in the experimental section.

**Preliminaries.** Let $q$ be a natural language query and $G = (V, E)$ be a directed graph representing input structured data. Each node $u \in V$ corresponds to a text chunk with an average token length $d$, and each directed edge $(u_1, u_2) \in E$ represents some structural dependence from a *source chunk* $u_1$ to a *target chunk* $u_2$. The objective is for an LLM $f$ to generate an answer $a = f(q, G)$ by encoding both $q$ and $G$. This task requires the LLM to comprehend the individual textual content of nodes while also properly leveraging and reasoning over the structural inductive biases encoded in $G$.

*Sequential Encoding* is the default approach in modern LLMs [42, 13, 23]. It involves processing an arbitrary linearized sequence of text chunks, denoted without loss of generality as $[u_1, u_2, \ldots, u_n]$, where each $u_i \in V$ is composed of a sequence of token embeddings, by using the concatenated sequence $[u_1, u_2, \ldots, u_n, q]$ as input. Modern LLMs commonly employ causal attention, where each token attends to all preceding tokens. This mechanism results in a computational complexity of $\mathcal{O}(n^2 L^2)$, where $L$ denotes the max chunk length. A consequence of sequential encoding is the sensitivity of the model output to input order, termed *positional biases*. Moreover, this default setting lacks inherent mechanisms for leveraging structural inductive biases, should they exist in the data.

*Parallel Text Encoding* is adopted by an alternative line of work [43, 61, 12, 35] that treats text chunks as an unordered set, $S = \{u_1, u_2, \ldots, u_n\}$. Here, the LLM encodes each chunk $u_i \in S$ independently and in parallel, often with all chunks sharing the same PEs to signify their lack of explicit order. The computational complexity for this encoding process is $\mathcal{O}(nL^2)$, linear in $n$ (the number of chunks). This method, however, discards the modeling of direct interactions among chunks, thereby sacrificing relational information critical for multi-hop reasoning in favor of efficiency. During answer generation, tokens attend to all encoded text chunks in $S$ and the query $q$.

Graph-KV injects structural inductive bias using two main strategies: a structure-aware attention mechanism and appropriately assigned shared PEs.

## 3.1 Graph-KV

**The Structure-aware Attention Mechanism.** First, Graph-KVperforms offline parallel encoding of each text chunk $u_i$ to obtain its initial latent representation $h_{u_i}^{(0)}$. These representations, $\{h_{u_i}^{(0)}\}_{u_i \in V}$, are then used to form initial Key-Value (KV) pairs, denoted as $\{(k_{u_i}^{(0)}, v_{u_i}^{(0)})\}_{u_i \in V}$, which can be loaded into the KV cache. Following the graph structure $G = (V, E)$, Graph-KV updates the representation of a target chunk $u_j$ by modeling its relationships via the attention mechanism with its source chunks, denoted as $\mathcal{N}(j) = \{u_i \mid (u_i, u_j) \in E\}$. This update is achieved by computing a sparsified attention:

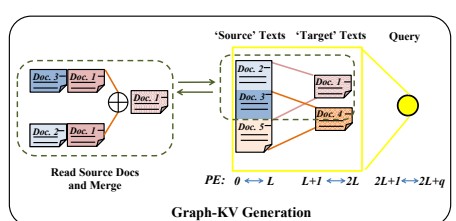

Figure 2: PE-sharing mechanism in Graph-KV. As shown on the **right side**, source docs share one PE range, while targets share another. Attending Doc.1 to the KVs of its sources (Doc.2 and Doc.3), is functionally equivalent to the **left side**: reading Doc.2 followed by Doc.1, and Doc.3 followed by Doc.1, then merging the resulting representations of Doc.1.

$\text{softmax}\left(\frac{Q_j K_{\mathcal{N}(j)}^\top}{\sqrt{d_h}}\right) V_{\mathcal{N}(j)}$ where $Q_j$ is the Query vector associated with $u_j$ (the target chunk), $K_{\mathcal{N}(j)} = [k_{u_i}^{(0)}]_{u_i \in \mathcal{N}(j)}$ and $V_{\mathcal{N}(j)} = [v_{u_i}^{(0)}]_{u_i \in \mathcal{N}(j)}$ are matrices formed by stacking the key-values of its source chunks, respectively, and $d_h$ used for normalization denotes the dimension of QK values.

This update procedure can be iterated for multiple rounds. However, for our experiments, we conduct a single round (i.e., $t = 1$). This serves as a proof-of-concept to model interactions extending beyond those addressed by existing parallel encoding methods, which typically process chunks as purely independent, while preserving low computational complexity. Iterating for multiple rounds does not yield significant performance gains on the current evaluation tasks. Nevertheless, we believe that with more complex tasks, or if the model were further fine-tuned to adapt to this new attention architecture, greater improvements could be anticipated. Finally, upon query, input query tokens and subsequently generated answer tokens attend to the representations of both source and target chunks.

**The Allocation of Positional Encodings.** Our allocation of PEs aims to reduce positional bias and improve context efficiency. We begin by considering the scenario without explicit directed edges, where graph-structured data effectively becomes a collection of independent text chunks. This configuration is analogous to those in many existing studies on parallel encoding [61, 43, 35]. To mitigate positional bias for these independent chunks, all such text chunks are assigned positions within the shared range $[0, L)$, where we assume a maximum chunk length of L. Furthermore, when structural dependencies are present in the data (e.g., via directed edges), target chunks are subsequently assigned positions from $L$ to $2L$, i.e., within another shared range $[L, 2L)$ immediately following the first. Query tokens and any generated tokens are then allocated positions in a range beyond 2L, subsequent to those of the target chunks.

While this study does not investigate the iterative application of target chunks as source chunks in subsequent processing rounds, the proposed methodology permits such natural extension. For instance, target chunks in a subsequent round could be allocated positions within the range $[2L, 3L)$, with query token positions adjusted correspondingly; this iterative pattern can be continued as needed. A key benefit of this procedure is its conservation of the context window: since numerous chunks share identical positional ranges, the overall required positional span is only about $TL$. Here, $T$ denotes the number of iterations (a generally small constant), and $L$ is often less than $10k$.

To illustrate how latent representations of target chunks are formed by Graph-KV, please refer to Fig. 2. Suppose there are directed edges doc.2 $\rightarrow$ doc.1 and doc.3 $\rightarrow$ doc.1, Graph-KV can be understood as guiding the LLM to process two effective "documents": one formed by doc.2 followed by doc.1, and another by doc.3 followed by doc.1. The representations of the doc.1 portions obtained from both these effective documents, are then aggregated. Consequently, the representation of target chunk doc.1 contains the information reflecting its connections to source chunks doc.2 and doc.3.

**Computational Complexity.** The representations for all text chunks in the first round are computed with complexity $\mathcal{O}(|V|L^2)$, aligning with previous parallel encoding methods. Suppose $\mathcal{T}$ is a set of target chunks, updating target chunk representations has a complexity of $\mathcal{O}(\sum_{u_j \in \mathcal{T}} |\mathcal{N}(j)|L^2) = \mathcal{O}(|E|L^2)$ as $|E|$ is the total number of such dependencies. During query time, the attention complexity for each query or generated token over all source and target chunks is $\mathcal{O}(|V|L)$, similar to vanilla sequential and parallel encoding schemes at generation.

**Remark: Attention Sink.** APE [61] considers sharing the PE range but avoids using the first several positions to avoid the attention sink problem [59]. We find that Graph-KV remains unaffected even though its chunks share the PE range from the first token. This is because we adopt the model that has been fine-tuned with independent attention blocks [35] to fit this change.

## 4   Experiments

**Task Selection** We design four tasks to evaluate Graph-KV, including three real-world applications and one stress test: **Task 1:** Retrieval-Augmented Generation (RAG), **Task 2:** ARXIV-QA, a new task of multi-hop paper understanding, **Task 3:** Paper topic classification, which is a classical graph learning task; **Task 4:** Stress test on scalability and efficiency over synthetic data.

**Backbone for Graph-KV.** Graph-KV necessitates the independent encoding of different text segments. This process introduces a distributional shift standard LLM backbones [43]. Two primary solutions address this challenge. The first involves applying a tuning-free heuristic, such as APE [61], which alleviates the shift by adjusting the temperature and scaling of attention weights. The second approach is to post-train the model with attention masks composed of independent attention blocks for different text chunks, such as Block-RAG [35]. Empirically, we found that the fine-tuned model exhibits more stable performance, particularly when employing Graph-KV, as demonstrated in subsequent experiments. Consequently, we default to using the `llama-3.1-8B-block-ft` (8B-Block) model as the backbone for Graph-KV. This model is based on the pre-trained `llama-3.1-8B` and is further post-trained with independent attention blocks [35] on the `tulu-3` dataset [28] and $20k$ RAG training instances from 2Wiki [17] and TriviaQA [22].

Due to limited computational resources, our work focuses on the `llama-3.1-8B` family, and we have not extended this specific tuning to other LLMs. However, our findings are, in principle, generalizable. Furthermore, we did not attempt to directly fine-tune the model with Graph-KV, although we believe such a step could further enhance its performance on many of the tasks discussed later.

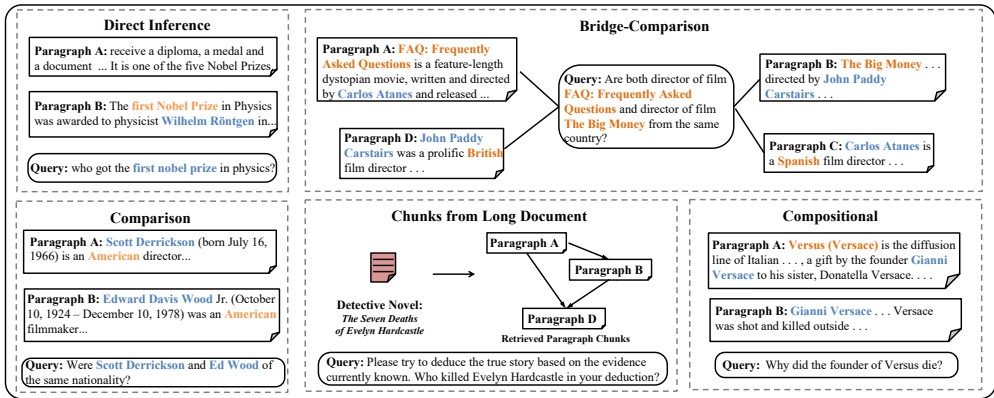

Figure 4: The reasoning settings in RAG tasks . *Direct inference* task requires identifying evidence chunks (from NarrativeQA [25]). Others that require multi-hop reasoning include multi-hop reasoning (*comparison*, *bridge* and *compositional* (from 2Wiki [17], HotpotQA [62]) and *long-document understanding* (from LongBench-v2 [4]). In these tasks, there exists implicit temporal or logical dependencies among the retrieved chunks.

**Baselines for comparison. 1) Sequential Encoding:** We consider two models which conducts serialized next-token prediction during post-training based on `llama-3.1-8B`. One is `llama-3.1-8B-sft` (8B-SFT) which is fully supervised tuned on the tulu-3 dataset. For fair comparison, we also take `llama-3.1-8B-rag` (8B-RAG), which is further tuned with the extra RAG data that is used for `llama-3.1-8B-block-ft` (8B-Block), which were also used for comparison in [35]. The two models encode inputs in a standard serialized manner, serving as direct baselines for Graph-KV, particularly in assessing its ability to leverage structural inductive biases, eliminate positional bias, and reduce context window consumption. **2) Parallel Context Window (PCW) [43]**, **3) Adaptive Parallel Encoding (APE) [61]** and **4) Block-RAG [35]** are methods with block attentions. Block-RAG serves as another direct baseline for Graph-KV as they adopt the same backbone LLM.

### 4.1 Task 1: Retrieval Augmented Generation (RAG)

Examples of the task scenarios in RAG are shown in Fig. 4, including direct inference, multi-hop reasoning, and long document understanding. We evaluate these scenarios using a total of 7 datasets, including NarrativeQA [25], TriviaQA [22], HotpotQA [62], 2Wiki [17], Multihop-RAG [49], MorehopQA [44] and LongBenchV2 [4]. For all the datasets, 10 text chunks are provided, and accuracy is selected as the primary metric. We follow [35, 33, 18] to judge whether any correct answers appear in the predicted output. See more implementation details in Appendix. A.2.1.

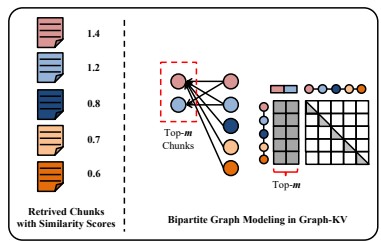

Figure 3: Graph-KV modeling for RAG.

**Graph-KV for Structure Modeling in RAG.** In RAG tasks, especially those involving multi-hop reasoning, text segments exhibit strong logical or temporal structural inductive biases. However, these dependencies are implicit and could not be cheaply modeled. We do not assume the dependencies are explicit, instead we construct them using a bipartite graph of text chunks. As illustrated in Fig. 3. Graph-KV tries to capture the structural dependencies among retrieved chunks without introducing too much complexity. Specifically, the retrieved chunks with the top-$m$ ($m$=1,3 in our experiments) similarity scores are considered source chunks, the KV-cache of them is concatenated, and attended by each of the remaining text chunks to independently generate the corresponding KV values. We also consider all retrieved chunks as both source and target chunks (named as "Graph-KV-Full"), which can be viewed as modeling all potential pair-wise dependencies.

**Result Analysis.** Experiment results on RAG datasets are displayed in Tables. 2 3, with key insights as follows: Across all the tasks, results show a clear trend: Graph-KV-Full generally outperforms its sparsified variants. Specifically, Graph-KV-Top-3 generally achieves better performance than $m$=1. Among them, the Graph-KV-Top-1 outperforms the Block-RAG [35] baseline except on Single Document QA while Graph-KV-Top-3 consistently outperforms Block-RAG across all cases, which reveals that our injected sparse dependency effectively complements the lack of dependencies among text chunks in parallel encoding methods. Notably, in tasks that emphasize multi-hop reasoning, or medium and long reasoning from long documents (Table. 2), Graph-KV-Top-3 significantly

| Backbone | Attention | MultiHop-RAG | | | | | MorehopQA | | | | | | Single Document QA in LongBench-V2 | | | | | |
|---|---|---|---|---|---|---|---|---|---|---|---|---|---|---|---|---|---|---|
| | | Infer | Compare | Temporal | Null | Avg | Hop-1 | Hop-2 | Hop-3 | Hop-4 | Hop-5 | Avg | Easy | Hard | Short | Medium | Long | Avg |
| | | 816 | 856 | 583 | 301 | 2556 | 444 | 416 | 154 | 13 | 91 | 1118 | 64 | 111 | 74 | 77 | 24 | 175 |
| 8B-SFT | Sequential | 42.54 | 33.49 | 33.37 | 99.34 | 39.18 | 57.65 | 24.27 | 29.87 | 15.38 | 26.37 | 38.37 | 35.9 | 28.8 | 35.1 | 31.2 | 20.8 | 31.4 |
| 8B-SFT | PCW | 69.39 | 34.29 | 34.35 | 42.10 | 42.01 | 11.26 | 10.81 | 10.38 | 0.00 | 17.58 | 11.35 | 8.3 | 20.0 | 15.4 | 20.0 | 0.0 | 15.6 |
| 8B-SFT | APE | 48.34 | 33.46 | 33.49 | 86.25 | 40.34 | 13.28 | 15.86 | 9.47 | 61.53 | 10.98 | 14.13 | 34.5 | 21.4 | 24.1 | 31.2 | 20.0 | 26.8 |
| 8B-RAG | Sequential | 73.12 | 34.66 | 39.00 | 43.43 | 44.27 | 54.72 | 22.59 | 22.07 | 53.85 | 25.27 | 35.86 | 32.8 | 32.4 | 35.1 | 35.1 | 16.7 | 32.6 |
| 8B-RAG | PCW | 50.37 | 33.62 | 34.19 | 48.16 | 39.35 | 0.22 | 1.20 | 0.00 | 0.00 | 1.09 | 0.62 | 16.7 | 27.1 | 29.2 | 21.9 | 9.5 | 23.3 |
| 8B-RAG | APE | 51.71 | 34.08 | 38.08 | 39.97 | 40.10 | 22.97 | 27.88 | 163.2 | 46.15 | 32.96 | 24.95 | 25.0 | 19.4 | 10.5 | 31.8 | 16.7 | 21.3 |
| 8B-Block | Sequential | 63.55 | 34.54 | 38.03 | 53.09 | 43.60 | 59.23 | 20.43 | 25.32 | 23.07 | 30.76 | 37.38 | 35.9 | 30.6 | 36.5 | 32.5 | 20.8 | 32.6 |
| 8B-Block | Block-RAG | 72.73 | 34.16 | 38.03 | 40.73 | 43.32 | 56.75 | 25.72 | 24.67 | 15.38 | 27.47 | 37.92 | 37.5 | 27.9 | 32.4 | 35.1 | 16.7 | 31.4 |
| 8B-Block | Graph-KV Top-1 | 73.25↑ | 34.19 | 38.18 | 53.85 | 44.81 | 54.72 | 25.72 | 24.67 | 23.07 | 31.86 | 37.96 | 34.9 | 27.0 | 32.9 | 28.6 | 25.0 | 29.9 |
| 8B-Block | Graph-KV Top-3 | 73.12 | 34.43 | 38.69 | 63.64 | 45.79 | 56.53 | 25.72 | 24.67 | 23.07 | 29.67 | 38.11 | 38.5↑ | 29.5 | 32.4 | 35.7↑ | 32.1 | 32.5 |
| 8B-Block | Graph-KV Full | 69.04 | 34.63 | 38.48 | 88.79 | 46.41↑ | 64.86↑ | 30.52↑ | 25.97 | 30.76 | 47.25↑ | 44.90↑ | 37.5 | 31.5↑ | 32.4 | 35.1 | 33.3↑ | 37.5↑ |

Table 2: Performance on Multihop-RAG [49], MorehopQA [44] and single documentQA in LongBench-V2 [4]. The best sequential encoding method is underlined, the best non-sequential approach is **bolded**. ↑ refers that the best non-sequential approach outperforms the best sequential encoding.

| Backbone | Attention | NarrativeQA | | 2Wiki | | | | TriviaQA | Hotpot QA | | |
|---|---|---|---|---|---|---|---|---|---|---|---|
| | | Infer | Compare | Infer | Bridge | Compose | Avg | Infer | Compare | Bridge | Avg |
| | | 3610 | 3040 | 1549 | 2751 | 5236 | 12576 | 11313 | 1487 | 5918 | 7405 |
| 8B-SFT | Sequential | 60.60 | 86.84 | 24.66 | 74.62 | 67.20 | 68.33 | 76.19 | 74.37 | 73.18 | 73.42 |
| 8B-SFT | PCW | 39.22 | 70.13 | 14.07 | 67.17 | 32.58 | 46.94 | 60.13 | 54.87 | 33.28 | 37.62 |
| 8B-SFT | APE | 49.05 | 74.44 | 18.26 | 61.17 | 37.41 | 49.20 | 66.28 | 60.52 | 46.24 | 49.11 |
| 8B-RAG | Sequential | 62.38 | 82.29 | 60.94 | 90.54 | 68.22 | 75.75 | 76.38 | 73.43 | 78.59 | 77.55 |
| 8B-RAG | PCW | 46.59 | 66.21 | 18.71 | 72.88 | 19.49 | 42.37 | 64.18 | 58.84 | 33.40 | 33.51 |
| 8B-RAG | APE | 49.14 | 74.40 | 21.88 | 84.55 | 29.96 | 51.65 | 67.31 | 63.34 | 42.41 | 46.61 |
| 8B-Block | Sequential | 63.21 | 84.24 | 63.08 | 90.07 | 70.76 | 77.29 | 76.64 | 71.88 | 78.82 | 77.43 |
| 8B-Block | Block-RAG | 59.39 | 83.25 | 50.41 | 89.85 | 60.90 | 71.35 | 74.44 | 70.20 | 71.59 | 71.31 |
| 8B-Block | Graph-KV Top-1 | 62.04 | 82.76 | 52.42 | 90.00 | 65.94 | 73.60 | 75.17 | 70.41 | 74.14 | 73.39 |
| 8B-Block | Graph-KV Top-3 | 62.29 | 83.25 | 56.74 | 90.47 | 67.83 | 75.15 | 75.66 | 71.01 | 76.24 | 75.19 |
| 8B-Block | Graph-KV Full | 62.88 | 82.69 | 54.16 | 90.43 | 67.11 | 74.38 | 75.85 | 70.41 | 77.62 | 76.17 |

Table 3: Performance on NarrativeQA [25], 2Wiki [17], TriviaQA [22] and HotpotQA [62]. The best sequential encoding method is underlined, the best non-sequential approach is **bolded**. ↑ refers that the best non-sequential approach outperforms the best sequential encoding.

outperforms Sequential Encoding by about $2\% - 10\%$. In the rest tasks (in Table. 3), although the gaps are generally narrower Graph-KV-Top-3 still outperforms Block-RAG by $4.65\%$ in *bridge* on HotpotQA, $6.93\%$ in *compose* on 2Wiki. Graph-KV-Top-3 only achieves performance comparable to, but not exceeding, the sequential baseline. This is because, in many tasks in Table. 3 such as *inference* and *comparison* (e.g. NarrativeQA [25], TriviaQA [22])), models can often directly infer from one text chunk. In the tasks that require two-hop reasoning, such as *bridge* or *compose*, sequential encoding may model structural dependencies that are originally held in the serialized order.

Regarding tuning-free approaches, although APE [61] consistently outperforms PCW [43] due to its temperature and scale adjustments, a substantial performance gap remains between the tuning-free approach APE and the fine-tuned model Block-RAG. This reveals the need for fine-tuning to enable LLMs that were pretrained on full causal attention to effectively encode text chunks independently and decode properly from these text chunks.

To evaluate the applicability of Graph-KV on general LLMs, we take Llama-3.1-8B-SFT as the backbone and provide detailed results in Appendix. A.1. While we observe a performance decline compared with the 8B-Block backbone due to distribution shift in the absence of post-training with block-wise attention, Graph-KV still outperforms the other parallel encoding baselines. Furthermore, as demonstrated in Appendix. A.1.2, employing multi-hop graph construction with Graph-KV yields additional performance improvements.

## 4.2 Task 2: ARXIV-QA – Multi-Hop Reasoning over Citation Network with Full Texts

**Experimental Setup.** We randomly selected 100 academic papers from the Arxiv dataset [20], each with its corresponding reference papers. Following a data processing and cleaning phase, which focused on the relationships between the primary paper and its references (detailed in Appendix A.2.2), we curated a final dataset comprising 60 primary papers. Including their references, this dataset encompassed a total of 472 papers with full-text availability. We then formulated one technical question for each of the 60 papers. Answering these questions necessitates: 1) inter-

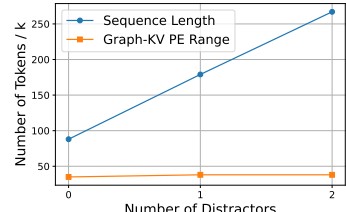

Figure 7: Average input sequence length (equivalent to position index range in sequential encoding) on ARXIV-QA with $0, 1, 2$ distractors, compared to the position index range used in Graph-KV.

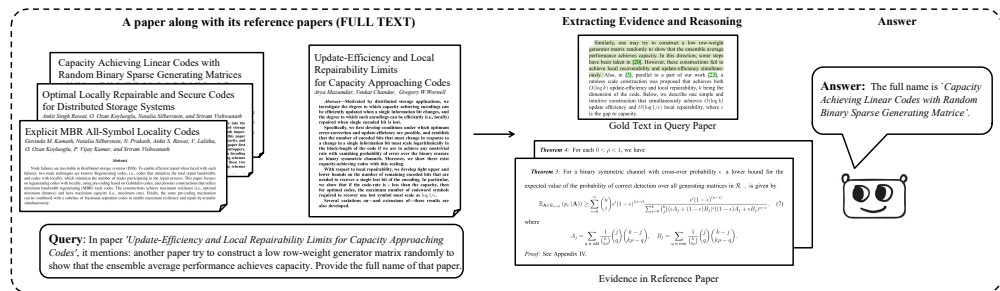

Figure 5: An example from the ARXIV-QA task. One needs first locate the central paper's introduction of the low row-weight generator matrix, and then compare the described methods with the content across all provided references (e.g., Theorems 1 and 3 in the ground-truth reference paper) to arrive at the correct answer.

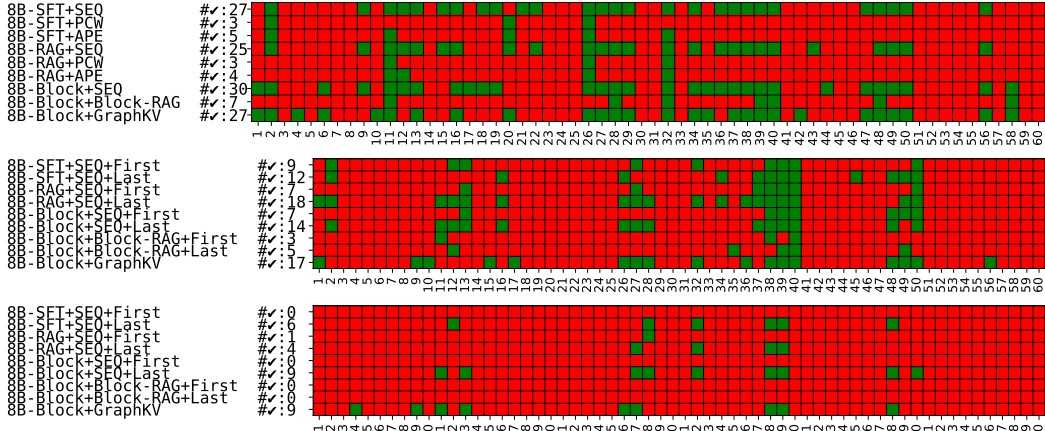

Figure 6: QA accuracy when querying with 0, 1, 2 distractors (**up to down**) on the 60 questions from ARXIV-QA. The number of correct answers is provided after ✓. 'SEQ' refers to sequential encoding. Green entry means correct answer and Red refers to wrong. When querying with distractors, 'Last' and 'First' refers to the position where the paper along with its references that are contain the answer are positioned (at the beginning or end of the sequence).

preting the main context of the paper, 2) comprehending the citation relationships, and 3) understanding the context of its references. An illustrative example is provided in Fig. 5 (note: the figure size has been significantly reduced due to space constraints; readers are encouraged to zoom in for detailed content). On average, addressing each question involves processing approximately $88k$ tokens of context, which approaches the $128k$ effective context window limit of the foundational models used.

To further elevate the difficulty of the question-answering task, we introduced distractors for each question. This was achieved by randomly grouping multiple papers along with their respective references. This expanded setup enables the evaluation of the model's positional bias (e.g., by varying the placement of the relevant paper and its references within the paper sequence) and rigorously tests the boundaries of extremely long contexts. For instance, grouping three papers and their references (i.e., one relevant paper plus two distractor papers, along with all their associated references) results in an average input length of 264.6k tokens. Fig. 7 provides a comparison.

**Graph-KV for Structure-Aware Modeling in ARXIV-QA.** Understanding technical details in a paper often involves reviewing its references, particularly when methods in the references are adapted to solving the problem in the current work. Graph-KV forms target-source pairs connecting the central paper with all its references. When distractors are included and text length surpasses 128k tokens, the shared PE strategy is vital for the LLM to properly digest the full context.

**Result Analysis.** Fig. 6 displays the QA accuracy. When queried solely with the paper and its references containing the answer (without distractors), sequential encoding demonstrates strong performance in this straightforward scenario. However, all parallel text encoding baselines (PCW, APE, and Block-RAG) fail to capture cross-document connections, resulting in significantly fewer correct answers. In the setup including distractors, sequential encoding exhibits severe positional bias. When the relevant paper is placed at the end of the sequence, performance remains comparable

to Graph-KV, primarily due to the recency bias of pre-trained auto-regressive LLMs, which tend to focus more on later-positioned text chunks [58]. Conversely, when relevant texts are positioned at the beginning of the sequence, the distractors and extended contexts lead to substantial performance degradation. For instance, all sequential encoding baselines fail to answer correctly when queried with two distractors. This demonstrates the limitations of sequential encoding in long-context, multi-source structured reasoning. Due to its PE sharing strategy and structural inductive bias injection, Graph-KV does not suffer from the positional bias and consistently achieves the best performance.

## 4.3 Paper Topic Classification

We further evaluated Graph-KV on the paper topic classification task using the Cora [36] and Pubmed [45] citation graphs. This task, which originated from graph learning as 'node classification', requires LLMs to classify a central paper into one of several categories based on its abstract, title, and neighbors. See Appendix. A.2.3 for details. Each central paper may have hundreds of references (e.g. up to 130 in Pubmed), thus making the task challenging. To verify the effectiveness of including reference information, we added a 'Query-Only' baseline that only feeds the model the central paper. For all methods, the central paper was consistently

| Backbone | Attention | Cora | Pubmed |
|---|---|---|---|
| 8B-SFT | Sequential | 66.66±0.62 | 80.64±0.39 |
| 8B-SFT | PCW | 68.63 | 76.95 |
| 8B-SFT | APE | 66.92 | 77.01 |
| 8B-RAG | Sequential | 70.35±0.17 | 82.06±0.16 |
| 8B-RAG | PCW | 66.05 | 76.52 |
| 8B-RAG | APE | 68.46 | 76.49 |
| 8B-Block | Query-Only | 57.38 | 83.60 |
| 8B-Block | Sequential | 67.09±0.17 | 79.79±0.16 |
| 8B-Block | Block-RAG | 69.55±0.30 | 83.24±1.16 |
| 8B-Block | Graph-KV | **71.03** | **84.61** |

Table 4: Performance on paper topic classification. Sequential encoding and Block-RAG produce varied answer due to different placement order of references. 'Query-Only' means only providing the central paper.

placed at the end of the sequence. Since Sequential encoding and Block-RAG showed varying performance depending on the order of references placed before the central paper, we report their average performance across seeds 42 to 44. In contrast, Graph-KV is robust to the order of references. The results, presented in Table 4, show that by incorporating the dependency on references from the central paper, Graph-KV outperforms both sequential encoding and parallel text encoding baselines. It is important to note that existing works applying LLMs to this traditional graph learning task typically either perform classification in the text embedding latent space [32, 8, 7] or train an adapter to map sampled reference papers into the LLM's token space [6, 48, 52]. Graph-KV is the first approach to address this task by adjusting the fundamental mechanism of LLMs.

## 4.4 Task 4: Stress Test on Scalability & Efficiency

We conduct stress test on synthetic data with an Nvidia RTX6000 GPU (48GB) with AMD EPYC 7763 64-core processor, to compare Graph-KV with sequential encoding baseline on scalability and efficiency. For details, refer to Appendix. A.2.4. 1) We construct synthetic star graphs (similar to the ego graphs used in previous tasks) with fixed word number on each node of 500 and 1000. We then gradually increase the number of neighbors to as-

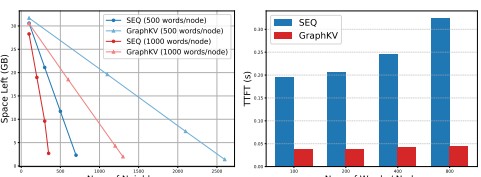

Figure 8: **Left:** Memory left w.r.t. num of nodes to encode. **Right:** TTFT w.r.t. num words per node.

sess GPU peak memory usage and report the remaining available memory. Results are presented on the left side of Fig. 8. Graph-KV is capable of encoding more than 3 times the number of neighbors compared with sequential encoding. 2) To evaluate efficiency, we report the time-to-first-token (TTFT) latency on star graphs with a fixed 10 neighbors, varying the number of words per node from 100 to 800. The results are shown on the right side of Fig. 8. The benefit stems from pre-filling text chunks with injected structural biases, a capability not achievable with sequential encoding.

## 5 Conclusion and Future Work

This paper introduces Graph-KV, a novel approach designed to overcome the limitations of auto-regressive LLMs in processing structured data. It achieves this by directly injecting structural inductive biases into the attention mechanism and employing strategic positional encoding, which in turn reduces positional bias and context window demands. Evaluations across diverse tasks, including RAG, a new academic QA benchmark (ARXIV-QA), and paper topic classification, demonstrate

Graph-KV's substantial outperformance of sequential and parallel encoding baselines, particularly in multi-hop reasoning and long-context scenarios. Although this work currently evaluates Graph-KV on one-hop structural dependencies, the core idea of leveraging structural dependencies to improve LLM understanding of more intricate data topologies holds significant promise for broader research.

## Acknowledgements

H. Wang, M. Li, S. Liu, S. Miao and P. Li are partially supported by NSF awards IIS-2239565, CCF-2402816, IIS-2428777, PHY-2117997; DOE award DE-FOA-0002785; JPMC faculty awards; Openai Research Credits; and Meta research award.

We extend our sincere gratitude to Xinyu Yang and Beidi Chen for their valuable discussion.

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

# A  Appendix

## A.1  Additional Experiment Results

### A.1.1  Applicability to General LLM

We apply Graph-KV to standard Llama-3.1-8B-SFT, comparing it against the parallel encoding baselines. Graph-KV consistently outperforms all parallel baselines, as shown in Table. 56.

| | MorehopQA | | | | | | 2Wiki | | | | |
|---|---|---|---|---|---|---|---|---|---|---|---|
| Attention | Hop-1 | Hop-2 | Hop-3 | Hop-4 | Hop-5 | Avg | Compare | Infer | Bridge | Compose | Avg |
| PCW | 11.26 | 10.81 | 10.38 | 0.00 | 17.58 | 11.35 | 70.13 | 13.07 | 67.17 | 32.58 | 46.94 |
| APE | 13.28 | 15.86 | 9.47 | **61.53** | 10.98 | 14.13 | 74.44 | **18.26** | 61.17 | 37.41 | 49.20 |
| Block-RAG | 13.96 | 13.94 | 9.74 | 0.00 | 15.38 | 13.32 | 72.17 | 15.17 | **68.01** | 35.54 | 48.99 |
| Graph-KV Top-3 | **15.54** | **16.10** | **12.98** | 0.00 | **18.68** | **15.47** | **76.41** | 17.43 | 64.15 | 47.19 | **54.30** |

Table 5: Results with Llama-3.1-8B-SFT.

| | Multihop-RAG | | | | | HotpotQA | | | NarrativeQA | TriviaQA |
|---|---|---|---|---|---|---|---|---|---|---|
| Attention | Infer | Compare | Temporal | Null | Avg | Compare | Bridge | Avg | Infer | Infer |
| PCW | 69.39 | 34.29 | 34.35 | 42.10 | 42.01 | 54.87 | 33.28 | 37.62 | 39.22 | 60.13 |
| APE | 48.34 | 33.46 | 33.49 | **86.25** | 40.34 | 60.52 | 46.24 | **49.11** | 49.05 | 66.28 |
| Block-RAG | 49.28 | 33.54 | 33.76 | 41.86 | 38.41 | 57.83 | 37.39 | 41.49 | **46.53** | 65.55 |
| Graph-KV Top-3 | **67.00** | **35.14** | **34.76** | 42.74 | **42.35** | **63.55** | **51.44** | 53.90 | 51.08 | 69.47 |

Table 6: Results with Llama-3.1-8B-SFT.

### A.1.2  Multi-Hop Graph-KV

We built multi-layer graphs from retriever relevance scores for iterative information propagation. The design of multi-hop propagation is as follows:

• 2-Hop: We design a chain of (Top 1 chunk) $\rightarrow$ (Top 2-4 chunks) $\rightarrow$ (Rest).

Iteration 1: "Top 1" acts as the source to update the KV of "Top 2-4".

Iteration 2: The updated "Top 2-4" then update the rest.

• 3-Hop: A deeper chain (Top 1) $\rightarrow$ (Top 2-3) $\rightarrow$ (Top 4-5) $\rightarrow$ (Rest) with three-step propagation.

| | | Multihop-RAG | | | | | HotpotQA | | | NarrativeQA | TriviaQA |
|---|---|---|---|---|---|---|---|---|---|---|---|
| Attention | #hop | Infer | Compare | Temporal | Null | Avg | Compare | Bridge | Avg | Infer | Infer |
| Graph-KV Top-1 | 1 | 73.25 | 34.19 | 38.18 | 53.85 | 44.81 | 70.41 | 74.14 | 73.39 | 62.04 | 75.17 |
| Graph-KV Top-3 | 1 | 73.12 | 34.43 | 38.69 | 63.64 | 45.79 | 71.01 | 76.24 | 75.19 | 62.29 | 75.66 |
| Graph-KV Full | 1 | 69.04 | 34.63 | 38.48 | **88.79** | 46.41 | 70.41 | **77.62** | 76.17 | **62.88** | 75.85 |
| Graph-KV Top-1-Top-3 | 2 | 73.20 | 34.54 | **38.78** | 70.82 | 46.15 | 71.95 | 77.25 | 76.19 | 62.60 | 75.90 |
| Graph-KV Top1-Top3-Top5 | 3 | **73.30** | **34.76** | **38.78** | 75.44 | **46.74** | **72.02** | 77.57 | **76.45** | 62.74 | **76.04** |

| | | MorehopQA | | | | | | 2Wiki | | | | |
|---|---|---|---|---|---|---|---|---|---|---|---|---|
| Attention | #hop | Hop-1 | Hop-2 | Hop-3 | Hop-4 | Hop-5 | Avg | Compare | Infer | Bridge | Compose | Avg |
| Graph-KV Top1 | 1 | 54.72 | 25.72 | 24.67 | 23.07 | 31.86 | 37.96 | 82.76 | 52.42 | 90.00 | 65.94 | 73.60 |
| Graph-KV Top3 | 1 | 56.63 | 25.72 | 24.67 | 23.07 | 29.67 | 38.11 | 83.25 | 56.74 | 90.47 | 67.83 | 75.15 |
| Graph_KV Full | 1 | 64.86 | 30.52 | 25.97 | 30.76 | **47.25** | 44.90 | 82.69 | 54.16 | 90.43 | 67.11 | 74.38 |
| Graph-KV Top1-Top3 | 2 | 67.11 | 32.69 | 25.32 | 30.76 | 37.36 | 45.70 | 83.45 | **57.04** | 90.77 | 68.97 | 75.76 |
| Graph-KV Top1-Top3-Top5 | 3 | **67.56** | **32.93** | **25.97** | **38.46** | 43.95 | **46.69** | **83.75** | **57.04** | 90.77 | **69.34** | **75.99** |

Table 7: Multi-Hop performance.

## A.2  Implementation Details

### Hardware and Platform

For all the experiments involved in this study, the code is implemented using PyTorch [37], the HuggingFace Transformers library [57], and FlashAttention-2 [9]. As to hardware, for the task ARXIV-QA, the parallel text encoding baselines (Block-RAG, PCW, APE) and Graph-KV run on 4 NVIDIA A100 Tensor Core GPUs, while the sequential encoding baseline runs on 8 NVIDIA A100

Tensor Core GPUs, as it requires higher memory. For the other tasks, all the methods run on with NVIDIA RTX 6000 Ada GPUs.

**Model Weights** For all the experiments, we adopt the open-source weight of `llama-3.1-8B-sft`[*], `llama-3.1-8B-TAG`[†] and `llama-3.1-8B-Block-FT`[‡] release by [35].

We do not further fine-tune the `llama-3.1-8B` with Graph-KV due to limited computational resources, although we believe that doing so could further improve performance on the experiments.

### A.2.1 Implementation Details for RAG

**Data Process and Evaluation** For 2Wiki [17], NarrativeQA [25], TriviaQA [22], MorehopQA [44] and HotpotQA [62], the data processing (the process to retrieve text chunk and the evaluation pipeline) strictly follows Block-RAG [35][§]. For Mulihop-RAG [49], the data processing and evaluation follows the original implementation[¶]. Across all benchmarks, the top 10 text chunks retrieved based on similarity scores are included in the input prompt in ascending order with respect to the scores [35]. Following [35, 33, 18], we use accuracy as the metric, and evaluate whether the correct answer appears in the output. For all methods, the output is constrained to a maximum of 256 tokens.

**Prompt:** For RAG tasks, the entire prompt input is divided into 3 parts, namely *Prefix*, *Text Chunks*, and *Question*, with each formatted as follows:

- *Prefix*: To ensure fair comparison, all the methods adopt the same prefix as follows:

```
You are an intelligent AI assistant. Please answer questions based on the user\'s
 instructions. Below are some reference documents that may help you in answering
the user\'s question.
```

- *Text Chunks*: The format for each text chunk (10 chunks in total for each question) is as follows:

```
-Title: {Title #1}. \n {Text #1}
-Title: {Title #2}. \n {Text #2}
...
-Title: {Title #10}. \n {Text #10}
```

- *Question*: All the methods adopt the same question format as follows:

For 2Wiki [17], HotpotQA [62], NarrativeQA [25] and TriviaQA [22], the question prompt follows those used in [35]:

```
Please write a high-quality answer for the given question using only the provided
 search documents (some of which might be irrelevant). \n Question: {Question}
```

For MultiHop-QA [49], we adopt the prompt from the original implementation, which is as follows:

```
Please write a high-quality answer for the given question using only the provided
 search documents. The answer to the question is a word or entity. If the
provided information is insufficient to answer the question, respond '
Insufficient Information'. Please finally give your answer started with: 'The
answer is:'. \n Question: {Question}
```

For MoreHop-QA [44], the prompt is also from the original implementation, which is:

```
Please write a high-quality answer for the given question using only the provided
 search documents. (If the answer is a date, format is as follows: YYYY-MM-DD (
ISO standard).) After thinking step by step, give your final answer following '
Answer:' \n Question: {Question}
```

---

[*]https://huggingface.co/ldsjmdy/Tulu3-SFT
[†]https://huggingface.co/ldsjmdy/Tulu3-RAG
[‡]https://huggingface.co/ldsjmdy/Tulu3-Block-FT
[§]https://github.com/TemporaryLoRA/Block-Attention
[¶]https://github.com/yixuantt/MultiHop-RAG

**Implementation**

- Sequential Encoding: Sequential method directly feeds the model with the sequence of *Prefix + Text Chunks + Question*.

- Parallel Encoding: independently encode the *Prefix*, each one of the *Text Chunks*, and *Question*, and then concatenate the KV cache together. The positional encoding setup follows the implementation used in the corresponding papers.

- Graph-KV: independently encode *Prefix* and *Question*, while inject the structural inductive biases as introduced in Section. 4.1.

### A.2.2  Implementation Details for ARXIV-QA

**Dataset Curation** We initially sample 100 central papers along with their references from the OGBN-ARXIV [20] citation network. Using the Arxiv API[||], we download the PDF files for each paper and convert them to full text using the `fitz` library[**]. During data cleaning, we ensure that the correct papers are downloaded and that each contains at least three valid references. For each reference, if it appears in a standalone sentence in the central paper—indicating that the central paper uses at least one sentence to compare or discuss the reference—we manually design a corresponding question. Through this process, 60 central papers with associated questions are selected to form the ARXIV-QA task, which is publicly available[††].

**Distractors** When adding distractors, we also employ randoms seeds $42 - 44$ to randomly sample distractors paper citation ego-graphs, but we do not observe different outputs across the methods with different seeds.

**Prompt:** For ARXIV-QA, the input prompt can be divided into 3 parts, namely *Prefix*, *Paper texts*, *Question*:

- *Prefix*: To ensure fair comparison, all the methods adopt the same prefix as follows:

```
You are an intelligent AI assistant. You will first read the related works of a
paper, then you will read the paper. Then answer the question.
```

- *Paper texts*:

```
{Full Text of Reference a}, {Full Text of Reference b},..., {Full Text of
Reference k} \n\n Now please read the paper: {Full Text of Central Paper #1}

(if with distractors:)
(Distractor #1){Full Text of Reference l}, {Full Text of Reference m},..., {Full
Text of Reference s} \n\n Now please read the paper: {Full Text of Central Paper
#2}

(Distractor #2){Full Text of Reference t}, {Full Text of Reference u},..., {Full
Text of Reference y} \n\n Now please read the paper: {Full Text of Central Paper
#3}
```

- *Question*: To ensure fair comparison, all the attention methods adopt the same question as follows:

```
Question: {Question}
```

**Implementation**

- Sequential Encoding: directly input the sequence of *Prefix + Paper Texts + Question*. When distractors are included, the *Paper Texts* are repeated accordingly, while ensuring that each citation ego-graph remains intact, as illustrated above.

---

[||]https://info.arxiv.org/help/api/index.html
[**]https://pymupdf.readthedocs.io/en/latest/tutorial.html
[††]See the zip file

• Parallel Encoding: independently encode the *Prefix*, the full text of each reference and each paper, *Question*, and then concatenate the KV cache together. The positional encoding setup follows the implementation used in the corresponding paper.

• Graph-KV: independently encode *Prefix* and *Question*, while inject the structural inductive biases following the method introduced in Section. 4.2.

**Demo Example** Below we display two examples of question of ARXIV-QA, and the corresponding answers generated by different attention methods. Full questions and ground truth answers could be found in[‡‡].

---



**Demo Example #1**

**Question:** In paper 'xLED: Covert Data Exfiltration from Air-Gapped Networks via Router LEDs', the authors describe a method from another paper that introduces a type of malware aimed bridging the air-gap between computers and a nearby mobile phone by exploiting FM radio signals emanating from the video card. You've also read this paper, please provide the full name of the paper.

**Ground Truth:** AirHopper: Bridging the Air-Gap between Isolated Networks and Mobile Phones using Radio Frequencies

**Without Distractor**

**8B-SFT+SEQ:** The paper that introduces a type of malware aimed bridging the air-gap between computers and a nearby mobile phone by exploiting FM radio signals emanating from the video card is titled *AirHopper: Bridging the Air-Gap between Isolated Networks and Mobile Phones using Radio Frequencies*.

**8B-RAG+SEQ:** The paper that introduces a type of malware aimed bridging the air-gap between computers and a nearby mobile phone by exploiting FM radio signals emanating from the video card is titled *AirHopper: Bridging the Air-Gap between Isolated Networks and Mobile Phones using Radio Frequencies*

**8B-Block-FT+SEQ:** The paper that introduces a type of malware aimed at bridging the air-gap between computers and a nearby mobile phone by exploiting FM radio signals emanating from the video card is titled *AirHopper: Bridging the Air-Gap between Isolated Networks and Mobile Phones using Radio Frequencies*

**8B-Block-FT+Block-RAG:** The paper that introduces a type of malware aimed at bridging the air-gap between computers and a nearby mobile phone by exploiting FM radio signals emanating from the video card is titled COTTONMOUTH: An Asymmetric Subtle Software Attack by Mordechai Guri, Gabi Kedma, Assaf Kachlon, and Yuval Elovici.

**8B-Block-FT+Graph-KV:** The paper that introduces a type of malware aimed bridging the air-gap between computers and a nearby mobile phone by exploiting FM radio signals emanating from the video card is titled: *AirHopper: Bridging the Air-Gap between Isolated Networks and Mobile Phones using Radio Frequencies*. Authors: Mordechai Guri, Gabi Kedma, Assaf Kachlon, Yuval Elovici.



---

[‡‡]zip file of supplementary material

**With #1 Distractor**

**8B-SFT+SEQ-First:** The full name of the paper is xLED: Covert Data Exfiltration from Air-Gapped Networks via Router LEDs.

**8B-SFT+SEQ-Last:** *AirHopper: Bridging the Air-Gap between Isolated Networks and Mobile Phones using Radio Frequencies*

**8B-RAG+SEQ-First:** According to the provided information, the paper that introduces the malware that exploits FM radio signals emanating from the video card to bridge the air-gap between computers and a nearby mobile phone is xLED: Covert Data Exfiltration from Air-Gapped Networks via Router LEDs

**8B-RAG+SEQ-Last:** According to the provided information, the paper you're referring to is AirHopper: *Bridging the Air-Gap between Isolated Networks and Mobile Phones using Radio Frequencies* by Mordechai Guri, Yosef Solewicz, Andrey Daidakulov, and Yuval Elovici.

**8B-Block-FT+SEQ-First:** The paper you are referring to is titled Software-Defined Radio and the RF Fingerprints of Computersby Mordechai Guri, Yuval Elovici, and Boris Zadov.

**8B-Block-FT+SEQ-Last:** The paper you are referring to is ÄirHopper: Bridging the Air-Gap between Isolated Networks and Mobile Phones using Radio Frequenciesby Mordechai Guri, K. Gabi, K. Assaf, and Yuval Elovici.

**8B-Block-FT+Block-RAG-First:** <end of text>

**8B-Block-FT+Block-RAG-Last:** The authors of the 5. The 5. The 5. and the 5. The 5. The 5. and the 5. The 5. The 5. The 5. The 5. The

**8B-Block-FT+Graph-KV:** The paper that introduces a type of malware aimed bridging the air-gap between computers and a nearby mobile phone by exploiting FM radio signals emanating from the video card is titled *AirHopper: Bridging the Air-Gap between Isolated Networks and Mobile Phones using Radio Frequencies* by Mordechai Guri, Gabi Kedma, Assaf Kachlon, and Yuval Elovici.

---

**With #2 Distractors**

**8B-SFT+SEQ-First:** The full name of the paper is Learning to Bridge the Air-Gap between Isolated Networks and Mobile Devices using RF Modulation from the Video Card

**8B-SFT+SEQ-Last:** xLED: Covert Data Exfiltration from Air-Gapped Networks via Router LEDs

**8B-RAG+SEQ-First:** <end of text>

**8B-RAG+SEQ-Last:** <end of text>

**8B-Block-FT+SEQ-First:** <end of text>

**8B-Block-FT+SEQ-Last:** <end of text>

**8B-Block-FT+Block-RAG-First:** Null

**8B-Block-FT+Block-RAG-Last:** Null

**8B-Block-FT+Graph-KV:** The paper that introduces a type of malware aimed bridging the air-gap between computers and a nearby mobile phone by exploiting FM radio signals emanating from the video card is: *AirHopper: Bridging the air-gap between isolated networks and mobile phones using radio frequencies* by Mordechai Guri, Gabi Kedma, Assaf Kachlon, and Yuval Elovici.

**Demo Example #2**

**Question:** In paper 'Fast Multi-frame Stereo Scene Flow with Motion Segmentation', they directly adopt an edge detector, which originates from another paper that you've read. Provide the full name of that paper.

**Ground Truth:** Fast Edge Detection Using Structured Forests

Without Distractor

**8B-SFT+SEQ:** The edge detector adopted in the paper 'Fast Multi-frame Stereo Scene Flow with Motion Segmentation' originates from the paper titled *Fast Edge Detection Using Structured Forests* by Piotr Dollár and C. Lawrence Zitnick.

**8B-RAG+SEQ:** The paper Fast Multi-frame Stereo Scene Flow with Motion Segmentation by Tatsunori Taniai, Sudipta N. Sinha, and Yoichi Sato directly adopts an edge detector from the paper *Fast Edge Detection Using Structured Forests* by Piotr Dollár and C. Lawrence Zitnick.

**8B-Block-FT+SEQ:** According to the paper Fast Multi-frame Stereo Scene Flow with Motion Segmentation by Tatsunori Taniai, Sudipta N. Sinha, and Yoichi Sato, they directly adopt an edge detector from the paper *Fast Edge Detection Using Structured Forests* by Piotr Dollár and C. Lawrence Zitnick.

**8B-Block-FT+Block-RAG:** The edge detector adopted in the paper Fast Multi-frame Stereo Scene Flow with Motion Segmentation is the Census Transform, which is a robust and efficient edge detection method. The Census Transform is a non-parametric image matching technique that was first introduced by Zabih and Woodfill in their paper Non-parametric Local Transforms for Computing Visual Correspondence (Zabih and Woodfill, 1994).

**8B-Block-FT+Graph-KV:** The paper Fast Multi-frame Stereo Scene Flow with Motion Segmentation directly adopts an edge detector that originates from the paper *Fast Edge Detection Using Structured Forests* by Piotr Dollár and C. Lawrence Zitnick.

**With #1 Distractor**

**8B-SFT+SEQ-First:** The edge detector adopted in the paper 'Fast Multi-frame Stereo Scene Flow with Motion Segmentation' is Dollár and Zitnick's *Fast Edge Detection Using Structured Forests* (DOI: 10.1109/TPAMI.2015.2467470).

**8B-SFT+SEQ-Last:** The edge detector adopted in the paper 'Fast Multi-frame Stereo Scene Flow with Motion Segmentation' is the *Fast Edge Detection Using Structured Forests* by Piotr Dollár and C. Lawrence Zitnick.

**8B-RAG+SEQ-First:** According to the supplementary material of the paper Fast Multi-frame Stereo Scene Flow with Motion Segmentation by Tatsunori Taniai, Sudipta N. Sinha, and Yoichi Sato, they directly adopt an edge detector from the paper *Fast Edge Detection Using Structured Forests* by Piotr Dollár and C. Lawrence Zitnick.

**8B-RAG+SEQ-Last:** According to the paper Fast Multi-frame Stereo Scene Flow with Motion Segmentation by Tatsunori Taniai, Sudipta N. Sinha, and Yoichi Sato, the edge detector they adopted is from the paper *Fast Edge Detection Using Structured Forests* by Piotr Dollár and C. Lawrence Zitnick.

**8B-Block-FT+SEQ-First:** According to the supplementary material of the paper Fast Multi-frame Stereo Scene Flow with Motion Segmentation by Tatsunori Taniai, Sudipta N. Sinha, and Yoichi Sato, the edge detector used in their work originates from the paper *Fast Edge Detection Using Structured Forests* by Piotr Dollár and C. Lawrence Zitnick.

**8B-Block-FT+SEQ-Last:** The paper Fast Multi-frame Stereo Scene Flow with Motion Segmentation directly adopts an edge detector from the paper *Fast Edge Detection Using Structured Forests* by Piotr Dollár and C. Lawrence Zitnick.

**8B-Block-FT+Block-RAG-First:** The paper that is directly adopted as an edge detector in 'Fast Multi-frame Stereo Scene Flow with Motion Segmentation' is *Fast Edge Detection Using Structured Forests* by Piotr Dollár and C. Lawrence Zitnick.

**8B-Block-FT+Block-RAG-Last:** The paper Fast Multi-frame Stereo Scene Flow with Motion Segmentation by Tatsunori Taniai, Sudipta N. Sinha, and Yoichi Sato does not directly adopt an edge detector from another paper. Instead, they use a fast edge detector proposed by Dollár and Zitnick (2017) in their work A Continuous Optimization Approach for Efficient and Accurate Scene Flow

**8B-Block-FT+Graph-KV:** The paper Fast Multi-frame Stereo Scene Flow with Motion Segmentation directly adopts an edge detector from the paper *Fast Edge Detection Using Structured Forests* by Piotr Dollár and C. Lawrence Zitnick.

### A.2.3 Implementation Details for Paper Topic Classification

**Dataset Curation** The dataset is originally from Cora [36] and Pubmed [45], we adopt the test set split adopted in [7]. For each paper, the input text consists of the title and abstract.

**Prompt:** For Paper Topic Classification, the input prompt can be divided into 3 parts, namely *Prefix*, *Paper Title & Abstracts*, *Question*:

• *Prefix*: To ensure fair comparison, all the methods adopt the same prefix as follows:

```
You are an intelligent AI assistant. You will first read a list of titles or
abstracts of papers cited by a central paper. Then, you will read the title or
abstract of the central paper itself. Finally, you will answer a question related
 to the central paper:
```

• *Paper Title & Abstracts*:

```
{Title & Abstract of Reference 1}, {Title & Abstract of Reference 2},..., {Title
& Abstract of Reference k} \n\n They are all cited by the following central paper:
 : {Title & Abstract of Central Paper}
```

• *Question*: To ensure fair comparison, all the attention methods adopt the same question as follows:

```
Classify the central paper into one of the following categories: {Classes}.
Provide your answer following 'Answer:'
```

### Implementation

For sequential encoding and Block-RAG [35], we report the average performance with seeds $42 - 44$ to randomly shuffle the placement order of sequence.

• Sequential Encoding: directly input the sequence of *Prefix + Paper Title & Abstract + Question*. When distractors are included, the *Paper Texts* are repeated accordingly, while ensuring that each citation ego-graph remains intact.

• Parallel Encoding The parallel encoding baselines independently encode the *Prefix*, each paper title & abstract, *Question*, and then concatenate the KV cache together. The positional encoding setup follows the implementation used in the corresponding paper.

• Graph-KV: independently encode *Prefix* and *Question*, while inject the structural inductive biases with *Paper Title & Abstract* following the modeling as introduced in Section. 4.3.

### A.2.4 Implementation Details for Stress Test

As introduced in the main text, we employ an Nvidia RTX6000 GPU (48GB) with AMD EPYC 7763 64-core processor for stress test. Specifically, for attention implementation, all the methods use FlashAttention2 [9]. The raw text is extracted from the first test sample of the Cora [36] dataset. To meet the specified input length requirements—500 and 1000 words for the memory test, and 100, 200, 400, and 800 words for the generation latency evaluation—we either repeat or truncate the original text accordingly.

• Memory test: To test the memory usage of each method, we gradually increase the number of neighbors of synthetic star graph, and use torch.cuda.reset_peak_memory_stats() and torch.cuda.max_memory_allocated() functions to monitor the peak GPU memory usage.

• Time-To-First-Token (TTFT): Similar to other parallel encoding baselines, Graph-KV also benefits from KV-cache pre-filling. With pre-filled KV-cache, Graph-KV achieves significantly lower TTFT (Time-To-First-Token) compared to sequential encoding. To measure TTFT, we use the time.time() function to record the elapsed time.

