# OpenReview forum: "Graph-KV: Breaking Sequence via Injecting Structural Biases into Large Language Models"
_NeurIPS.cc/2025/Conference — NeurIPS 2025 poster_

### Official Review · Reviewer_mq6i · 2025-06-19

**Clarity:** 2
**Significance:** 2
**Originality:** 3
**Rating:** 4
**Confidence:** 3

**Summary:**

To inject structural inductive bias into auto-regressive models, this paper proposes Graph-KV, which leverages the KV-cache of text segments as condensed representations and governs their interaction through structural inductive biases. The experiments on seven RAG benchmarks, ARXIV-QA, and paper topic classification show that Graph-KV outperforms the baselines.

**Questions:**

See questions in “Strengths And Weaknesses”, here are some additional suggestions:
    • Figure 1, line 3, lack a space between “[66]” and “(2)”.
    • Line 153, lack a space between “Graph-KV” and “performs”.

**Ethical Concerns:**

["NO or VERY MINOR ethics concerns only"]

**Final Justification:**

I raised my score from 3 to 4 because the authors addressed the weaknesses. I did not further increase my score because the experimental results are not consistently strong.

**Limitations:**

yes

**Quality:**

2

**Strengths And Weaknesses:**

Strengths
    • The paper proposes Graph-KV to inject structural inductive bias into auto-regressive models.
    • Graph-KV exhibits better performances than other methods across three different scenarios.

Weaknesses
    • Some technical details are not clear. See the below questions.
Line 156-157, how is the representation $h$ used to form initial KV pairs?
Line 164-165, how is sparsified attention applied? It is not obvious in the self-attention equation.
In addition to the (K, V) pairs of the source chunk, is it also possible to consider (K, V) of the target chunk while using the equation at line 165?
What do the numbers of the “Backbone” rows in Table 2, 3 stand for?
    • Some experiments are missing.
Line 168-175, the benefits of multiple rounds are discussed, but there is no experiment supporting this.
While the Graph-KV is composed of 1) Structure-aware Attention Mechanism and 2) Allocation of Positional Encodings, there is no ablation study to discuss the effectiveness of the individual modules of Graph-KV.
    • The paper is not well organized. The implementation of Graph-KV across different scenarios and their results should be in different sections.

---

> ### Author Rebuttal · Authors · 2025-07-31
>
> We thank reviewer mq6i for their detailed review and suggestions. Below we try to address their questions.
>
> # [Q.1 How is the representation used in Initial KV pairs formulation]
> Below we try to explain how to use the representation in initial KV pairs formulation:
>
>
> The initial Key-Value (KV) pairs are generated from token embeddings using the standard mechanism within a Transformer's self-attention layer.
>
> The process is as follows:
>
> 1. **Tokenization and Embedding:** First, each text chunk is tokenized, and these tokens are mapped to their corresponding input embeddings. This sequence of embeddings is the latent representation h mentioned in our paper.
> 2. **Linear Projection:** Next, these embeddings (h) are processed by a self-attention layer. Inside this layer, two distinct linear projection matrices—a Key projector and a Value projector—transform the input embeddings into Key (K) and Value (V) vectors.
>
> These generated K and V vectors form the initial KV pairs, which are then stored in the KV cache for that text chunk.
>
>
> # [Q.2 Implementation of Sparsified Attention]
> Thank you for the question. We are happy to clarify the implementation of the sparsified attention mechanism.
>
> **How Attention is Sparsified**
>
> In a standard self-attention mechanism, a query attends to all key-value pairs in its context. Our approach is "sparsified" because a target chunk is restricted to attending only to a predefined subset of chunks: its designated sources.
>
> The equation on line 165 reflects this. For a given target chunk j, its query vectors (Qj​) are computed. However, the key (K) and value (V) matrices used in the attention calculation are constructed **exclusively** from the KV pairs of its source chunks, denoted as N(j). The KV pairs from all other non-source chunks are effectively masked out of the computation. This is analogous to message passing in graphs, where information propagates only along existing edges (from sources to the target).
>
> # [Q.3 Multiple Round Iteration of Graph-KV]
> We thank you for this constructive suggestion. Our initial 1-hop experiments have shown significant improvements across the three tasks. That said, we agree that evaluating performance on deeper, multi-hop dependencies is a critical test.
>
> In response, we conducted new experiments to directly assess Graph-KV's multi-hop capabilities. The results confirm that iterative updates improve performance.
>
> ## [Scenario 1] New Multi-Hop RAG Experiments
> To simulate deeper reasoning, we constructed multi-layered dependency graphs based on retriever relevance scores. Information is propagated through iterative Graph-KV updates:
>
> - 2-Hop Structure: We defined a dependency chain as **(Top 1 Chunk) → (Top 2-4 Chunks) → (Remaining Chunks)**.
>
> Iteration 1: The "Top 1" chunk acts as the source to update the KV-caches of the "Top 2-4" chunks.
>
> Iteration 2: The updated "Top 2-4" chunks then serve as the source to update the remaining chunks.
>
> - 3-Hop Structure: We modeled an even deeper chain as **(Top 1) → (Top 2-3) → (Top 4-5) → (Remaining Chunks)**, realizing a three-step propagation process.
>
> |  |  | Multihop-RAG |  |  |  |  | HotpotQA |  |  | NarrativeQA | TriviaQA |
> |:---:|:---:|:---:|:---:|:---:|:---:|:---:|:---:|:---:|:---:|:---:|:---:|
> | Attention | #hop | Infer | Compare | Temporal | Null | Avg | Compare | Bridge | Avg | Infer | Infer |
> | Graph-KV Top-1 | 1 | 73.25 | 34.19 | 38.18 | 53.85 | 44.81 | 70.41 | 74.14 | 73.39 | 62.04 | 75.17 |
> | Graph-KV Top-3 | 1 | 73.12 | 34.43 | 38.69 | 63.64 | 45.79 | 71.01 | 76.24 | 75.19 | 62.29 | 75.66 |
> | Graph-KV Full | 1 | 69.04 | 34.63 | 38.48 | **88.79** | 46.41 | 70.41 | 77.62 | 76.17 | **62.88** | 75.85 |
> | Graph-KV Top-1-Top-3 | 2 | 73.20 | 34.54 | **38.78** | 70.82 | 46.15 | 71.95 | 77.25 | 76.19 | 62.60 | 75.90 |
> | Graph-KV Top1-Top3-Top5 | 3 | **73.30** | **34.76** | **38.78** | 75.44 | **46.74** | **72.02** | **77.57** | **76.45** | 62.74 | **76.04** |
>
>
>
>
> |  |  | MorehopQA |  |  |  |  |  | 2Wiki |  |  |  |  |
> |:---:|:---:|:---:|:---:|:---:|:---:|:---:|:---:|:---:|:---:|:---:|:---:|:---:|
> | Attention | #hop | Hop-1 | Hop-2 | Hop-3 | Hop-4 | Hop-5 | Avg | Compare | Infer | Bridge | Compose | Avg |
> | Graph-KV Top1 | 1 | 54.72 | 25.72 | 24.67 | 23.07 | 31.86 | 37.96 | 82.76 | 52.42 | 90.00 | 65.94 | 73.60 |
> | Graph-KV Top3 | 1 | 56.63 | 25.72 | 24.67 | 23.07 | 29.67 | 38.11 | 83.25 | 56.74 | 90.47 | 67.83 | 75.15 |
> | Graph_KV Full | 1 | 64.86 | 30.52 | 25.97 | 30.76 | **47.25** | 44.90 | 82.69 | 54.16 | 90.43 | 67.11 | 74.38 |
> | Graph-KV Top1-Top3 | 2 | 67.11 | 32.69 | 25.32 | 30.76 | 37.36 | 45.70 | 83.45 | **57.04** | **90.77** | 68.97 | 75.76 |
> | Graph-KV Top1-Top3-Top5 | 3 | **67.56** | **32.93** | **25.97** | **38.46** | 43.95 | **46.69** | **83.75** | **57.04** | **90.77** | **69.34** | **75.99** |
>
>
>
> The results show a consistent trend: modeling deeper **2-hop and 3-hop** dependencies yields further performance gains over the strong 1-hop variants across most RAG benchmarks. This confirms that Graph-KV effectively models complex, multi-hop information flow and is not limited to single-step interactions.
>
>
> ## [Scenario 2] 2-Hop Paper Topic Classification
>
> We extend the paper topic classification experiment to a 2-hop citation graph: **(2-hop references) → (1-hop references) → (central paper)**. This creates a natural 2-iteration "message passing" scenario. To manage the graph size, we sampled k (k=3,10) references for each 1-hop neighbor.
>
> Incorporating information from these 2-hop neighbors via a second Graph-KV iteration further boosts classification accuracy.
>
> | Topic Classification | # samples | Cora | Pubmed |
> |:---:|:---:|:---:|:---:|
> | Graph-KV | -- | 71.03 | 84.61 |
> | Graph-KV (2Hop) | 3 | 71.21 | 85.04 |
> | Graph-KV (2Hop) | 10 | **71.77** | **85.32** |
>
>
>
> ## [conclusion]
> These new experiments validate that the Graph-KV framework is not limited to single-hop interactions but is a flexible method capable of modeling deeper, multi-hop dependencies when present in the data structure. Thank you for pushing us to explore this valuable direction.
>
> # [Ablation]
> Thank you for this excellent suggestion. We have performed the requested ablation study. Before presenting the results, it is important to clarify the deep synergy between Graph-KV's two core components, as they are designed to work together in principle.
>
> The **Structure-aware Attention** mechanism operates on a set of source documents that ideally should be treated without regard to their order (i.e., maintaining permutation equivariance). Using this mechanism without our **Shared Positional Encoding (PE)** would require imposing an arbitrary sequence on the source chunks. This would break the desired permutation equivariance and introduce severe positional bias, as the model's output would change based on the random ordering of the sources.
>
> Our Shared PE allocation is specifically designed to solve this problem by assigning the same positional information to all source chunks, making the model robust to their ordering. The following ablation study empirically validates this synergy.
>
> We tested four configurations on the topic classification task. Note that the variance (±) reflects sensitivity to the input order of reference documents, indicating positional bias.
>
> |  |  |  | Cora | Pubmed |
> |:---:|:---:|:---:|:---:|:---:|
> | Structural Attention | Shared PE |  |  |  |
> | No | No | Block-RAG | 69.55±0.30 | 83.24±1.16 |
> | No | Yes | -- | 69.74 | 83.59 |
> | Yes | No | -- | 70.14±0.69 | 83.75±1.02 |
> | Yes | Yes | Graph-KV | 71.03 | 84.61 |
>
> This study confirms our reasoning:
>
> - **Structural Attention alone:** As predicted, this configuration suffers from high positional bias (e.g., ±0.69 variance on Cora). Lacking permutation equivariance, its performance is unstable, though still better than the baseline.
>
> - **Shared PE alone:** This removes the positional bias but provides only a marginal performance gain, as it lacks the structure-aware mechanism to guide reasoning.
>
> - **Full Graph-KV:** By combining both components, our full Graph-KV method achieves the best of both worlds. It leverages structural dependencies for a significant performance boost and uses Shared PE to maintain permutation equivariance, resulting in the highest accuracy and stable performance.
>
> # [W. Organization, in different sections]
> Thank you for your feedback on the paper's organization. We apologize if the structure was not clear. We want to clarify that we have structured the experiments (Section 4) by task, with the implementation details for Graph-KV presented within each respective task's section.
>
> To clarify the existing layout:
>
> - **Task 1: Retrieval-Augmented Generation (RAG)**: The implementation of Graph-KV for RAG tasks is detailed in **Section 4.1** (lines 249-258), illustrated by Figure 3.
>
> - **Task 2: ARXIV-QA:** The setup and implementation for the ARXIV-QA task are presented in **Section 4.2** (lines 302-307), with an example in Figure 5.
>
> - **Task 3: Paper Topic Classification:** The modeling approach for topic classification is described in **Section 4.3** (lines 322-330) .
>
> To address your concern directly, we will improve the organization in the final version by adding more distinct sub-headings within each task's section (e.g., "Graph-KV Implementation" and "Result Analysis") to make the structure more explicit and easier to follow.
>
> # [Typos]
> Thanks for pointing out the typos, we will fix them in our final version.

---

> > ### Comment · Reviewer_mq6i · 2025-08-04
> > **acknowledgment**
> >
> > Thanks a lot for your response! I've raised my score to 4. I am still skeptical because the new experimental results are not consistently strong.

---

> > > ### Author Response · Authors · 2025-08-08
> > >
> > > We sincerely thank the reviewer for their continued engagement and for acknowledging our paper and rebuttal. We are grateful for the feedback on our new multi-hop message passing experiments. The reviewer observed some "inconsistency" in the multi-hop results. Below we try to give some further explanations:
> > >
> > > 1. As shown in Table. 2,3 in the paper, among methods that utilize parallel encoding or key-value (KV) manipulation, our sparsified single-hop Graph-KV Top-1 and Graph-KV Top-3, already achieve strong performance.
> > >
> > > 2. On the new experiment results, on some datasets, the sparsified Graph-KV with multi-hop iterations still performs slightly worse than Graph-KV Full. We believe this is because Graph-KV Full models all 1-hop pairwise dependencies, capturing an even bigger set of relationships than the multi-hop sparsified version. This comprehensive approach, while computationally more expensive, can sometimes be critical for achieving the highest performance.
> > >
> > > 3. Despite the above, the multi-hop approach consistently shows a gradual and consistent improvement in performance when compared to the single-hop sparsified methods (e.g., Top-1 or Top-3). This demonstrates its ability to progressively enrich information and enhance performance without incurring the full computational cost of Graph-KV Full.
> > >
> > > We thank the reviewer again for their time and valuable contributions to the discussion.

---

### Official Review · Reviewer_C8Lc · 2025-06-29

**Clarity:** 3
**Significance:** 3
**Originality:** 4
**Rating:** 5
**Confidence:** 3

**Summary:**

The paper introduces Graph-KV, a framework that injects structural biases into LLMs by modeling documents as graphs. Instead of sequential attention, target segments attend only to selected source segments based on graph-defined dependencies. Shared positional encodings reduce bias and context usage. Evaluated on RAG, academic QA, and citation classification tasks, Graph-KV outperforms sequential and parallel baselines, particularly in multi-hop reasoning and long-context settings.

**Questions:**

Further questions to discuss. Do the graphs' properties influence the model predictions? For example, if the graphs are dense, does the current framework still work well? If this is a dense graph, there should be many documents to merge.

**Ethical Concerns:**

["NO or VERY MINOR ethics concerns only"]

**Final Justification:**

The authors have addressed all of my concerns. I believe this approach of injecting chunks into LLMs is novel and represents exactly the right direction for integrating graphs into LLMs. I will maintain my positive score.

**Limitations:**

The paper does not discuss its limitations. However, a key concern is the scope of graph structures the framework can support and whether the fixed window length may pose constraints on its scalability.

**Quality:**

3

**Strengths And Weaknesses:**

The paper proposes a novel framework for modeling documents as graph structures and demonstrates its efficiency. The idea of encoding structural information through positional embeddings is particularly impressive and presents a promising direction for incorporating topological bias into LLMs. However, the paper’s main weakness lies in its unclear writing. Below are specific concerns.

It is unclear how the graph structure is built within the Graph-KV framework. Are the edges in the graph generated automatically? If automatically, what criteria or model is used to define source–target dependencies?

The framework uses position encoding windows of length L. How is this value chosen? Is it task-specific, empirically tuned, or theoretically motivated?

The relationship between the left and right sides of Figure 2 is confusing. What exactly does the “merge” operation refer to? There is no introduction in the corresponding paragraph.

typo in Table 2. wrong bold in MultiHop-8B-RAG,  Temporal.

---

> ### Author Rebuttal · Authors · 2025-07-31
>
> We thank **Reviewer C8Lc** for their insightful questions, constructive suggestions, and acknowledgment of Graph-KV's novelty and effectiveness. We address their questions below.
>
>
> # [Q.1 Edge Generation]
> Thank you for this insightful question. Yes, the graph edges are generated automatically based on the nature of the task. We consider two primary scenarios:
> 1. For Tasks Without an Explicit Graph (e.g., General RAG)
>
> In these cases, a graph structure is generated based on a relevance-based heuristic. The chunks with the top-k similarity scores to the query are designated as "source" nodes, and the rest are "targets." This models the intuition that understanding the most relevant documents is crucial for contextualizing the others. This is in contrast to the naive sequential models, which assume every pair of documents is connected and thus adopt dense attention.
> 2. For Tasks With a Natural Graph Structure (e.g., ARXIV-QA, Topic Classification)
>
> For these tasks, we use the pre-existing graph structure directly. The edges are automatically defined by the citation links between papers. This approach directly models the natural human reasoning process of consulting references for foundational knowledge.
>
> # [Q.2 Choose L]
> Thank you for the excellent question about how the PE window size, L, is determined. This value is not a fixed hyperparameter that requires tuning. Instead, it is determined dynamically and automatically for each input query.
>
> For any given input, L is set to the maximum token length found across all text chunks being processed for that query. This data-driven approach guarantees the shared PE windows are large enough to accommodate every chunk without truncation.
>
> Our framework then assigns these PE windows to different groups. As detailed in the paper, "source" chunks are assigned the shared PE range [0, L), and "target" chunks are assigned the subsequent shared range [L, 2L). Defining L as the maximum length of any single chunk ensures that all positional indices fall neatly within their designated windows.
>
> # [Q.3 Merge]
>
> We apologize that the term "merge" was not clearly defined and thank the reviewer for pointing this out. The term is intended as an analogy to describe how a target chunk synthesizes information from multiple source chunks within the attention mechanism.
>
> Figure 2 and its caption provide the intuition behind this analogy. For a target chunk (e.g.,
> Doc.1) with multiple source chunks (e.g., Doc.2 and Doc.3), the process is functionally equivalent to conceptually processing the sequence [Doc.2, Doc.1], then [Doc.3, Doc.1], and finally "merging" the resulting representations of Doc.1 from both processes.
>
> Technically, this "merge" is not a separate, explicit step. It is performed implicitly within a single, sparsified attention operation. The attention mechanism's output is a weighted sum of the value vectors from all sources, which naturally creates a single, updated representation for the target chunk that has effectively "merged" the contextual information from all its dependencies in one computational step.
>
> We will add this explicit clarification to the final version of the paper to improve readability.
>
>
> # [Q.4 Properties and Limitation]
>
> Thank you for this insightful question regarding the framework's scalability to dense graphs.
>
> You are correct that a dense graph implies a large number of documents, and our framework is well-suited for this challenge. The key is the **shared PE strategy**, which is designed to handle a large number of documents without suffering the context window limitations of sequential models.
>
> Our **ARXIV-QA experiment** serves as a proof-of-concept for this. In this task, the model successfully processed extremely long inputs (up to 264.6k tokens) by using shared PEs. We note in the paper that this approach was "vital for the LLM to properly digest the full context", demonstrating its effectiveness in scenarios where traditional methods would fail.
> Therefore, the shared PE mechanism provides a robust and scalable solution for dense graphs, directly addressing one of the most significant challenges in long-context reasoning.
>
> # [Typo]
> Thanks for pointing out the typo, we will fix them in our final version.

---

> > ### Comment · Reviewer_C8Lc · 2025-08-05
> >
> > Thank you for the detailed response. The authors have addressed all of my concerns. I believe this approach of injecting chunks into LLMs is novel and represents exactly the right direction for integrating graphs into LLMs. I will maintain my positive score.

---

> > > ### Author Response · Authors · 2025-08-08
> > >
> > > We are deeply grateful for the reviewer's highly positive feedback and recognition of our paper. We are hopeful that our proposed method might contribute meaningfully to the community's efforts to explore new ways of encoding graph structures into large language models.

---

### Official Review · Reviewer_khbB · 2025-07-01

**Clarity:** 2
**Significance:** 3
**Originality:** 2
**Rating:** 4
**Confidence:** 3

**Summary:**

This paper addresses a limitation of autoregressive Large Language Models (LLMs): their inability to natively process structured data, forcing all inputs to be serialized into flat sequences.
The authors propose Graph-KV, which directly injects structural inductive bias into the LLM's attention mechanism during inference.
Specifically, target text chunks selectively attend only to the KV cache of their designated source chunks. This is complemented by strategically assigning shared positional encodings (PE), where all source chunks share one PE range (0-L), all target chunks share another PE range (L-2L), and finally the query is appended.
 Experiments across four different tasks on multiple datasets demonstrate the model's effectiveness.

**Questions:**

Q1. In the baselines for comparison, Graph-KV is applied to the llama-3.1-8B-block model. Would similar effects be achieved when applied to other LLM models (such as 8B-SFT or 8B-RAG)? The llama-3.1-8B-block itself has already been trained with block attention.

Q2. Can you provide preliminary experimental results or deeper analysis on multi-round iteration performance (e.g., t=2 or t=3)? Multi-hop message passing is common in graph neural network message passing patterns. For example, you could also compare computational overhead between Graph-KV Top-1/3 and Graph-KV Full.

Q3. Compared to the significant effects in Table 2, Graph-KV+8B-Block in Table 3 datasets performs worse than 8B-Block Sequential (though better than block RAG). While the authors acknowledge that Graph-KV doesn't outperform sequential encoding in certain reasoning tasks (e.g., NarrativeQA, TriviaQA), their explanation—that sequential structure already captures implicit dependencies—seems insufficiently validated. Given that the paper's core contribution is that structure-aware attention mechanisms can improve LLM performance, deeper analysis of the potential losses from Graph-KV sparse attention is necessary.

Q4. Regarding Table 2, it is unclear why all experimental groups and control groups show accuracy that first decreases then increases as hop-k increases.

**Ethical Concerns:**

["NO or VERY MINOR ethics concerns only"]

**Final Justification:**

I appreciate your responses and additional results. The new experiments address key concerns and strengthen the paper. While some limitations remain (e.g., unified edge weights), the overall contribution is solid, particularly for complex reasoning tasks.

**Limitations:**

Yes.

**Quality:**

2

**Strengths And Weaknesses:**

**Strengths:**

1. Graph-KV requires no additional parameters or complex prompting, instead modifying the LLM attention mechanism itself.

2. Thorough baseline comparisons and good computational efficiency.

3. By assigning new PE to preserve structural information of source and target chunks, positional bias is mitigated.

**Weaknesses:**

1. The method is based on block attention post-training on specific backbone models (such as llama-3.1-8B-block-ft), raising questions about applicability to other general LLMs.

2. Experiments mainly focus on single-hop dependencies (source → target). The authors mention that update steps can be iterated with t>1 to simulate deeper interactions, but note this doesn't bring significant improvement on current tasks. This may indicate limitations of the current method for more complex multi-hop graph topologies. Multi-hop graph topologies may have higher value. In the experimental section, on MorehopQA multi-hop answering, Graph-KV Top-1/3 performs significantly worse than Graph-KV Full, not demonstrating the superiority shown on other datasets.

3. In Table 3 datasets, experimental results show that 8B-Block Sequential performs better, suggesting that reasoning dependencies between two paragraphs may already be implicit in the sequential structure. Injecting structural bias through Graph-KV may not provide additional benefits or may even be counterproductive.

4. The paper lacks alternative evaluation methods for assessing chunk relevance. Are there scenarios where similarity scores cannot fully capture the relationships?

5. Although the authors mention that single-layer performance is already good and cite computational constraints, the lack of multi-layer/complex structure testing for comparison is not very convincing. Additionally, it seems this model does not reflect the role of edge weights (since top-k segments are directly concatenated together, there is no difference shown among these k segments).

6. Figure 6 explanation is unclear and difficult to understand.

---

> ### Author Rebuttal · Authors · 2025-07-31
>
> # [W.1 and Q.1: Applicability to General LLM]
> We clarify our motivation and **present new results** showing model-agnostic benefits.
> 1. Motivation. We deliberately used a specialized backbone for a robust evaluation. While Graph-KV is model-agnostic, applying KV-cache manipulation to standard LLM can cause **distributional shift**[1-3]. Solutions include: 1) Tuning-free heuristics(APE[1]), 2) Post-training on attention blocks (Block-RAG[3]). We chose post-training as it is more stable and state-of-the-art (SOTA), which best isolated Graph-KV's benefits.
> 2. Experiments on a Standard LLM. We **apply Graph-KV to standard Llama-3.1-8B-SFT**, comparing it against the parallel encoding baselines on the same backbone. **Graph-KV consistently outperforms all parallel baselines**.
>
> |  | MorehopQA |  |  |  |  |  | 2Wiki |  |  |  |  |
> |:---:|:---:|:---:|:---:|:---:|:---:|:---:|:---:|:---:|:---:|:---:|:---:|
> | Attention | Hop-1 | Hop-2 | Hop-3 | Hop-4 | Hop-5 | Avg | Compare | Infer | Bridge | Compose | Avg |
> | PCW | 11.26 | 10.81 | 10.38 | 0.00 | 17.58 | 11.35 | 70.13 | 13.07 | 67.17 | 32.58 | 46.94 |
> | APE | 13.28 | 15.86 | 9.47 | **61.53** | 10.98 | 14.13 | 74.44 | 18.26 | 61.17 | 37.41 | 49.20 |
> | Block-RAG | 13.96 | 13.94 | 9.74 | 0.00 | 15.38 | 13.32 | 72.17 | 15.17 | 68.01 | 35.54 | 48.99 |
> | Graph-KV Top-3 | **15.54** | **16.10** | **12.98** | 0.00 | **18.68** | **15.47** | 76.41 | 17.43 | 64.15 | 47.19 | 54.30 |
>
> |  | Multihop-RAG |  |  |  |  | HotpotQA |  |  | NarrativeQA | TriviaQA |
> |:---:|:---:|:---:|:---:|:---:|:---:|:---:|:---:|:---:|:---:|:---:|
> | Attention | Infer | Compare | Temporal | Null | Avg | Compare | Bridge | Avg | Infer | Infer |
> | PCW | **69.39** | 34.29 | 34.35 | 42.10 | 42.01 | 54.87 | 33.28 | 37.62 | 39.22 | 60.13 |
> | APE | 48.34 | 33.46 | 33.49 | 86.25 | 40.34 | 60.52 | 46.24 | 49.11 | 49.05 | 66.28 |
> | Block-RAG | 49.28 | 33.54 | 33.76 | 41.86 | 38.41 | 57.83 | 37.39 | 41.49 | 46.53 | 65.55 |
> | Graph-KV Top-3 | 67.00 | **35.14** | **34.76** | **42.74** | **42.35** | **63.55** | **51.44** | **53.90** | **51.08** | **69.47** |
>
> Even without a specialized backbone, Graph-KV shows clear advantages. We plan to fine-tune another model with block attention for the final version; this was infeasible during rebuttal.
>
> [1]APE: Faster and Longer Context-Augmented Generation via Adaptive Parallel Encoding.
> Yang, et al. ICLR 2025.
>
> [2] Parallel Context Windows for Large Language Models. Ratner, et al.
>
> [3] Block-Attention for Efficient Prefilling. Ma, et al. ICLR 2025.
> # [W.2, Q.2 Multi-hop topologies]
> We agree evaluating multi-hop dependencies is critical. We ran new experiments on Graph-KV's multi-hop ability, and results confirm iterative updates improve performance:
>
> ## [Scenario 1] Multi-Hop RAG
> We built multi-layer graphs from retriever relevance scores for iterative information propagation.
> - 2-Hop: We design a chain of **(Top 1 chunk) → (Top 2-4 chunks) → (Rest)**.
>
> Iteration 1: "Top 1" acts as the source to update the KV of "Top 2-4".
>
> Iteration 2: The updated "Top 2-4" then update the rest.
> - 3-Hop: A deeper chain **(Top 1) → (Top 2-3) → (Top 4-5) → (Rest)** with three-step propagation.
>
> |  |  | Multihop-RAG |  |  |  |  | HotpotQA |  |  | NarrativeQA | TriviaQA |
> |:---:|:---:|:---:|:---:|:---:|:---:|:---:|:---:|:---:|:---:|:---:|:---:|
> | Attention | #hop | Infer | Compare | Temporal | Null | Avg | Compare | Bridge | Avg | Infer | Infer |
> | Graph-KV Top-1 | 1 | 73.25 | 34.19 | 38.18 | 53.85 | 44.81 | 70.41 | 74.14 | 73.39 | 62.04 | 75.17 |
> | Graph-KV Top-3 | 1 | 73.12 | 34.43 | 38.69 | 63.64 | 45.79 | 71.01 | 76.24 | 75.19 | 62.29 | 75.66 |
> | Graph-KV Full | 1 | 69.04 | 34.63 | 38.48 | **88.79** | 46.41 | 70.41 | 77.62 | 76.17 | **62.88** | 75.85 |
> | Graph-KV Top-1-Top-3 | 2 | 73.20 | 34.54 | **38.78** | 70.82 | 46.15 | 71.95 | 77.25 | 76.19 | 62.60 | 75.90 |
> | Graph-KV Top1-Top3-Top5 | 3 | **73.30** | **34.76** | **38.78** | 75.44 | **46.74** | **72.02** | **77.57** | **76.45** | 62.74 | **76.04** |
>
> |  |  | MorehopQA |  |  |  |  |  | 2Wiki |  |  |  |  |
> |:---:|:---:|:---:|:---:|:---:|:---:|:---:|:---:|:---:|:---:|:---:|:---:|:---:|
> | Attention | #hop | Hop-1 | Hop-2 | Hop-3 | Hop-4 | Hop-5 | Avg | Compare | Infer | Bridge | Compose | Avg |
> | Graph-KV Top1 | 1 | 54.72 | 25.72 | 24.67 | 23.07 | 31.86 | 37.96 | 82.76 | 52.42 | 90.00 | 65.94 | 73.60 |
> | Graph-KV Top3 | 1 | 56.63 | 25.72 | 24.67 | 23.07 | 29.67 | 38.11 | 83.25 | 56.74 | 90.47 | 67.83 | 75.15 |
> | Graph_KV Full | 1 | 64.86 | 30.52 | 25.97 | 30.76 | **47.25** | 44.90 | 82.69 | 54.16 | 90.43 | 67.11 | 74.38 |
> | Graph-KV Top1-Top3 | 2 | 67.11 | 32.69 | 25.32 | 30.76 | 37.36 | 45.70 | 83.45 | **57.04** | **90.77** | 68.97 | 75.76 |
> | Graph-KV Top1-Top3-Top5 | 3 | **67.56** | **32.93** | **25.97** | **38.46** | 43.95 | **46.69** | **83.75** | **57.04** | **90.77** | **69.34** | **75.99** |
>
> ## [Scenario 2] 2-Hop Paper Topic Classification
>
> We extended to 2-hops via a (2-hop refs) → (1-hop refs) → (paper) message-passing chain. To manage graph size, we sampled k=3,10 references per 1-hop neighbor.
>
> | Topic Classification | # samples | Cora | Pubmed |
> |:---:|:---:|:---:|:---:|
> | Graph-KV | -- | 71.03 | 84.61 |
> | Graph-KV (2Hop) | 3 | 71.21 | 85.04 |
> | Graph-KV (2Hop) | 10 | **71.77** | **85.32** |
>
> ## [conclusion]
> These results validate that Graph-KV can model deeper dependencies beyond single-hop interactions. We thank you for pushing us in this valuable direction.
> # [W.3 Q.3 Concerns about Performance]
> We appreciate the detailed analysis. Here, we contextualize these nuances by highlighting Graph-KV's fundamental advantages.
> 1. Significant Gains on Complex Reasoning
>
> **Graph-KV consistently excels on complex reasoning benchmarks.**
> - Advanced Multi-Hop RAG (Table 2): On challenging multi-hop datasets, Graph-KV significantly surpasses all baselines, including a strong sequential model.
> - Long-Context Reasoning (ARXIV-QA, Figure 6):Only Graph-KV beats sequential models that are crippled by positional bias.
> - Classic Graph Learning (Table 4):It again outperforms all baselines.
>
> When structure is paramount, Graph-KV is a clear winner.
>
> 2. Understanding the Nuances when simpler (Table 3)
>
> On less complex tasks, the sequential model is comparable or slightly better. The benefit of Graph-KV is most pronounced on tasks where explicit, injected structure is more valuable than the implicit, linear order of documents. For simple tasks where the answer is in a single doc, vanilla attention can suffice. Graph-KV is designed for challenging cases.
>
> 3. Fundamental Advantages
>
> Graph-KV solves fundamental auto-regressive flaws, not just achieve high scores:
> - **Eliminates Positional Bias**: It is robust to document order, ensuring reliable reasoning.
> - **Far More Efficient**: Its reduced complexity enables reasoning over document scales infeasible for sequential models.
> - **Superior in its Class**: It consistently beats peers (e.g., Block-RAG) by modeling inter-document structure.
>
> 4. The Untapped Potential of Structure-Aware Training
>
> Sequential models have massive pre-training. Graph-KV achieves strong results without large-scale, structure-aware pre-training, yet it already rivals or surpasses the sequential baseline on complex tasks. We believe structure-aware fine-tuning would further improve its performance.
> # [W. 4 similarity scores cannot fully capture the relationships]
> We agree simple similarity scores are imperfect proxy to model structural dependencies. Our heuristic is a pragmatic choice for RAG tasks, where, as noted in the paper, true dependencies are implicit. We build the graph with the highest-scoring doc as the "source," based on the intuition that the most relevant chunks are often critical hubs for information flow in multi-hop reasoning. Though a heuristic, experiments prove this is highly effective, validating that an approximate structure is far better than none.
> # [W. 5 Edge weights]
> An excellent suggestion. We currently treat all dependencies (edges) as uniform, as our graphs, like many in practice, do not have explicit edge weights. However, we agree incorporating them is a compelling future direction. A straightforward extension is to modulate attention scores between nodes, allowing the model to prioritize important connections.
> # [W. 6 explanation of Figure. 6]
> Figure 6 shows model performance on ARXIV-QA with very long contexts and "distractor" docs. It illustrates standard models' positional bias issue and Graph-KV's robustness.
>
> In the figure, each grid shows a model's performance (Green=correct, Red=fail). The three panels (top to bottom) correspond to adding 0, 1, and 2 distractor docs, respectively. Adding distractors makes the context much longer and tests models to find information in a "noisy" environment .
> ## [Top Panel - 0 Distractor]
> Without distractors, Sequential models perform well, and Parallel methods perform poorly as they can't model cross-document connections. Graph-KV also performs well, nearly matching the top sequential performance by correctly using the citation graph structure..
> ## [Middle & Bottom Panel - Positional Bias Test with 1,2 Distractors]
> This is the most important part. For sequential models, relevant docs were placed either at the context's very beginning ("First") or very end ("Last").
> - **Sequential Models Fail**: Performance collapses when documents are placed "First" but succeeds when "Last" (due to recency bias). With two distractors, they get "lost in the middle."
> - **Graph-KV Remains Robust**: In contrast, its performance is high and stable regardless of document order or distractors because it uses explicit graph structure, not linear position.
> # [Q.4 Table. 2]
> This is likely an artifact for two reasons. **Data Scarcity:** The test set for 4- and 5-hop questions is too small for the accuracy scores to be statistically significant. **Inconsistent Complexity:** Question difficulty may not scale linearly with the hop count, as some higher-hop questions can be incidentally simpler.

---

> > ### Comment · Reviewer_khbB · 2025-08-05
> >
> > Thank you for your detailed and thoughtful rebuttal. I have reviewed your responses and additional experiments. Below are my updated views on the main concerns:
> >
> > **[W1 / Q1: Applicability to General LLM]**
> >
> > The newly added experiments applying Graph-KV to unified standard LLMs are valuable and clearly demonstrate the generalizability of your approach across models. This effectively addresses concerns about the method's applicability beyond post-trained block attention backbone models. I recommend including these results in the next version.
> >
> > **[W2 / Q2: Multi-hop topologies]**
> >
> > The additional experiments on deeper multi-hop structures, particularly the performance improvements shown on MorehopQA, are convincing. They support the claim that Graph-KV can handle deeper graph topologies. I suggest integrating these multi-hop findings into the main paper in the next version. Additionally, could the authors compare the computational time between Graph-KV Top-3, Graph-KV Full, and Graph-KV Top1-Top3-Top5? Since efficiency is also a major contribution of the method, and improvements on individual datasets are only around 1%, computational efficiency is one of the practical evaluation criteria.
> >
> > **[W3 / Q3: Concerns about Performance]**
> >
> > Your response provides useful clarification on when Graph-KV is most beneficial. The performance gains in complex multi-hop or long-context tasks, while advantages may be less apparent in simple tasks with strong sequential cues, is reasonable. This contextualization makes the evaluation more balanced and helps properly position when to use Graph-KV.
> >
> > **[W4: similarity scores cannot fully capture the relationships]**
> >
> > While similarity scores are heuristic, I agree they provide a concise and effective approach for many RAG scenarios. Based on the experiments, the method appears to produce useful structures. Your explanation supports its usage.
> >
> > **[W5: Edge weights]**
> >
> > As you acknowledge, the current model does not yet utilize edge weights. Further exploration of this in future work would be a valuable extension.
> >
> > **[W6: explanation of Figure 6]**
> >
> > Your clarification makes the purpose and design of the figure clearer.
> >
> > Overall:
> > I appreciate your responses and additional results. The new experiments address key concerns and strengthen the paper. While some limitations remain (e.g., unified edge weights), the overall contribution is solid, particularly for complex reasoning tasks. I am raising my score to 4.

---

> ### Author Response · Authors · 2025-08-08
>
> We thank the reviewer again for their insightful comments, which have pushed us to verify the general applicability of Graph-KV and its capability for multi-hop iteration. We will incorporate experimental results for these two aspects into the next version of our paper. We also appreciate the suggestion to discuss the complexity of multi-hop iterative Graph-KV. Below, we provide a preliminary analysis of the computational complexity, which will be expanded upon in the revised manuscript.
>
> As discussed in lines 199–204 of our paper, the complexity of sequential encoding is $O(∣V∣^2L^2)$, where $V$ is the number of text chunks and $L$ is the average number of tokens per chunk. The computational complexity of a single Graph-KV iteration is $O(∣E∣L^2)$, where E represents the number of dependencies. For a two-step, multi-hop iterative propagation, the complexity becomes $O((∣E_1​∣+∣E_2​∣)L^2)$, where $E_1​$ and $E_2$ ​are the numbers of dependencies in the first and second iterations, respectively.
> In real-world scenarios, the number of dependencies $(∣E∣)$ is often much smaller than the number of nodes $(∣V∣)$, making the computational complexity of these propagations sparse.
>
> We now analyze the complexity of building dependencies within the RAG task:
> - **Graph-KV Full:** A fully connected bipartite graph is constructed, so the number of dependencies, $∣E∣$, is on the order of $∣V∣^2$. The complexity is therefore comparable to that of sequential encoding.
> - **Graph-KV Top-3:** The number of dependencies is significantly reduced, requiring only $(3×∣V∣)L^2$ computations.
> - **Graph-KV Top-1-Top-3-Top-5:** The computational cost is $(5∣V∣+18)L^2$, which is still more efficient than Graph-KV Full.
>
> We will include a more detailed discussion and formal analysis in the next version of our paper to provide a comprehensive view of the complexity of our proposed method.
>
> We thank the reviewer once again for their time and valuable contributions to the discussion. Their active participation has greatly helped us refine our analysis and improve the paper.

---

### Decision · Program_Chairs · 2025-09-17

**Decision:**

Accept (poster)

**Comment:**

This paper proposes a way to incorporate various forms of structured data into LLM by modifying the attention masks and positional encodings. Evaluations are broad, and reviewers broadly agree on significance and execution of the paper are up-to-standard. I agree with them, and think the paper will be broadly interesting to the broader NeurIPS community. Therefore, I recommend acceptance.